# Scaling Ensemble Distribution Distillation to Many Classes with Proxy Targets

**Max Ryabinin**[*]
Yandex, HSE University
Moscow, Russia
mryabinin0@gmail.com

**Andrey Malinin** [*]
Yandex, HSE University
Moscow, Russia
am969@yandex-team.ru

**Mark Gales**
Cambridge-ALTA Institute
Cambridge, United Kingdom
mjfg@eng.cam.ac.uk

## Abstract

Ensembles of machine learning models yield improved system performance as well as robust and interpretable uncertainty estimates; however, their inference costs can be prohibitively high. *Ensemble Distribution Distillation* (EnD$^2$) is an approach that allows a single model to efficiently capture both the predictive performance and uncertainty estimates of an ensemble. For classification, this is achieved by training a Dirichlet distribution over the ensemble members' output distributions via the maximum likelihood criterion. Although theoretically principled, this work shows that the criterion exhibits poor convergence when applied to large-scale tasks where the number of classes is very high. Specifically, we show that for the Dirichlet log-likelihood criterion classes with low probability induce larger gradients than high-probability classes. Hence during training the model focuses on the distribution of the ensemble tail-class probabilities rather than the probability of the correct and closely related classes. We propose a new training objective which minimizes the reverse KL-divergence to a *Proxy-Dirichlet* target derived from the ensemble. This loss resolves the gradient issues of EnD$^2$, as we demonstrate both theoretically and empirically on the ImageNet, LibriSpeech, and WMT17 En-De datasets containing 1000, 5000, and 40,000 classes, respectively.

## 1 Introduction

As machine learning models are being deployed ever more widely into production, especially in safety-critical applications such as self-driving cars or medical diagnostics, obtaining reliable estimates of uncertainty in their predictions becomes increasingly important to guaranteeing their safety and reliability [1]. Ensemble methods are known to yield both improved predictive performance and reliable uncertainty estimates [2, 3, 4, 5, 6]. Furthermore, ensembles allow separating out *total uncertainty* into *data* and *knowledge uncertainty*[2]. The former is the intrinsic uncertainty due to class overlap and noise inherent in the data, while the latter is the model's uncertainty due to lack of understanding of the test data [7]. Estimates of *knowledge uncertainty* are often used to detect anomalous and unfamiliar inputs [8, 9, 10, 7] as well as in active learning [11].

Despite their numerous advantages, using ensembles can be computationally prohibitive. Obtaining predictions in real time is often expensive even for a single model, and the hardware requirements for serving an ensemble scale linearly with size. As a result, over the past several years the area of ensemble distillation has gained increasing attention. Broadly speaking, distillation methods aim to train a single model which can approximate the behavior of an ensemble, but at low computational cost. In the simplest and most frequently used form of distillation [12], the student model is trained to capture the average prediction of the ensemble: for example, in case of classification this reduces

---

[*]Equal contribution, order decided by coin toss.
[2]Data and Knowledge Uncertainty are also known as Aleatoric and Epistemic uncertainty.

35th Conference on Neural Information Processing Systems (NeurIPS 2021).

to minimizing the KL-divergence between the model and the ensemble mean. While this method allows the student to obtain predictive performance comparable to that of the original ensemble, the information about its distributional properties (in other words, its diversity) is lost in the process. As ensemble-based uncertainty estimation utilizes the information about disagreement between its members, such distillation approaches fail to preserve valuable information from the ensemble.

Recently, several works have proposed distillation procedures that capture information about both the mean and distribution of ensemble predictions within a single model [13, 14, 15, 16]. These can be broadly split into methods where the student has multiple output heads - one per ensemble member [14, 15], and those which yield a higher-order distribution over the ensemble's outputs [13, 16]. The former are conceptually simpler, but more computationally expensive and may model the ensemble in excessive detail; the latter approaches are more light-weight, but also complex and, as we show in this work, suffer from optimization issues. This latter class of approaches, which we refer to as *Ensemble Distribution Distillation* (EnD$^2$), is the focus of this work.

Ensemble Distribution Distillation views outputs of each ensemble member as samples from a higher-order Dirichlet or Normal-Wishart distribution, and the student model learns the parameters of that distribution. Typically, EnD$^2$ is done by maximizing the likelihood of the ensemble's output distributions under the Dirichlet or Normal-Wishart prior. While this criterion is theoretically sound, we show that for large-scale classification tasks with many classes, gradient-based optimization of it is highly problematic, which limits its usefulness in real-life production scenarios. In this work, we investigate the poor convergence of models trained with Ensemble Distribution Distillation at scale. We analyze the the Dirichlet log-likelihood criterion and show that it leads to high gradient norm values that affect optimization. Specifically, if the ensemble members' output distributions have most probability mass allocated to a few classes, with the remainder spread among a long tail of exponentially less likely classes, then the gradients associated with the tail-classes will be significantly larger than those associated with high-probability classes. As a result, the optimization procedure focuses on modelling this distribution of tail-class probabilities, preventing convergence.

To solve this, we propose a two-step solution. First, we transform the empirical distribution of ensemble member predictions into a *Proxy-target* Dirichlet distribution with the same statistics and to use it as the target during distribution distillation. Second, we show that it is crucial to minimize the *reverse* KL-divergence between the model and the Proxy-Dirichlet, as minimization of the *forward* KL-divergence exacerbates the optimization issues. The proposed training procedure allows the model to converge, mitigating the issue of gradient explosion. We demonstrate this by distribution-distilling ensembles of models trained on the ImageNet classification, LibriSpeech speech recognition and WMT17 English-German language translation datasets, where there are 1000, 5000 and classes, respectively. To accomplish the latter two, we extend EnD$^2$ to autoregressive models and refer to this approach as Sequence Ensemble Distribution Distillation (SenD$^2$). On both datasets the distribution-distilled models outperform the baselines and yield uncertainty estimates competitive this those of the original ensemble.

Thus, our contributions are as follows:

- We draw attention to and analyze the convergence issues of Dirichlet likelihood optimization when applied to tasks with large numbers of classes;
- We propose an alternative two-step training approach for Ensemble Distribution Distillation which overcomes the aforementioned optimization issues;
- We adapt *Ensemble Distribution Distillation* to auto-regressive models and propose Sequence Ensemble-Distribution Distillation (SEnD$^2$).

## 2   Preliminaries: Ensembles and Distillation

We view ensembles within a Bayesian framework, where the model parameters $\boldsymbol{\theta}$ are random variables over which a prior $p(\boldsymbol{\theta})$ is placed. The posterior distribution $p(\boldsymbol{\theta}|\mathcal{D})$ is obtained via Bayes' rule:

$$p(\boldsymbol{\theta}|\mathcal{D}) = \frac{p(\mathcal{D}|\boldsymbol{\theta})p(\boldsymbol{\theta})}{p(\mathcal{D})} \propto p(\mathcal{D}|\boldsymbol{\theta})p(\boldsymbol{\theta}) \tag{1}$$

Consider an ensemble of models $\{P(y|\boldsymbol{x}^*, \boldsymbol{\theta}^{(m)})\}_{m=1}^M$ sampled from the posterior:

$$\big\{P(y|\boldsymbol{x}, \boldsymbol{\theta}^{(m)})\big\}_{m=1}^M \rightarrow \big\{P(y|\boldsymbol{\pi}^{(m)})\big\}_{m=1}^M, \quad \boldsymbol{\pi}^{(m)} = \boldsymbol{f}(\boldsymbol{x}; \boldsymbol{\theta}^{(m)}), \ \boldsymbol{\theta}^{(m)} \sim p(\boldsymbol{\theta}|\mathcal{D}) \tag{2}$$

where $\boldsymbol{\pi}$ are the parameters of a categorical distribution $[\mathrm{P}(y = \omega_1), \cdots, \mathrm{P}(y = \omega_K)]^{\mathsf{T}}$. The predictive distribution, or *predictive posterior*, for a test input $\boldsymbol{x}^*$ is obtained by marginalization:

$$\mathrm{P}(y|\boldsymbol{x}^*, \mathcal{D}) = \mathbb{E}_{\mathrm{p}(\boldsymbol{\theta}|\mathcal{D})}\big[\mathrm{P}(y|\boldsymbol{x}^*, \boldsymbol{\theta})\big] \approx \frac{1}{M}\sum_{m=1}^{M}\mathrm{P}(y|\boldsymbol{x}^*, \boldsymbol{\theta}^{(m)}) \tag{3}$$

In practice this is intractable and we approximate via Monte-Carlo sampling. Given the ensemble, the entropy of the predictive posterior is a measure of *total uncertainty*. *Knowledge uncertainty* can be assessed via measures of the spread, or 'disagreement', of the ensemble such as *mutual information*:

$$\underbrace{\mathcal{I}[y, \boldsymbol{\theta}|\boldsymbol{x}^*, \mathcal{D}]}_{\text{Knowledge Uncertainty}} = \underbrace{\mathcal{H}\big[\mathbb{E}_{\mathrm{p}(\boldsymbol{\theta}|\mathcal{D})}[\mathrm{P}(y|\boldsymbol{x}^*, \boldsymbol{\theta})]\big]}_{\text{Total Uncertainty}} - \underbrace{\mathbb{E}_{\mathrm{p}(\boldsymbol{\theta}|\mathcal{D})}\big[\mathcal{H}[\mathrm{P}(y|\boldsymbol{x}^*, \boldsymbol{\theta})]\big]}_{\text{Expected Data Uncertainty}} \tag{4}$$

An alternative measure of spread, which is particularly useful for ensemble of autoregressive sequence models, is the *reverse-mutual information* (RMI) [17]:

$$\mathcal{M}\big[\boldsymbol{y}, \boldsymbol{\theta}|\boldsymbol{x}, \mathcal{D}\big] = \mathbb{E}_{\mathrm{p}(\boldsymbol{\theta}|\mathcal{D})}\Big[\mathbb{E}_{\mathrm{P}(\boldsymbol{y}|\boldsymbol{x}, \mathcal{D})}\Big[\ln\frac{\mathrm{P}(y|\boldsymbol{x}, \mathcal{D})}{\mathrm{P}(y|\boldsymbol{x}, \boldsymbol{\theta})}\Big]\Big] \tag{5}$$

While ensembles yield improved predictive performance and meaningful uncertainty estimates, they are computationally expensive. To reduce inference-time cost, an ensemble can be *distilled* into a single model $\mathrm{P}(y|\boldsymbol{x}; \boldsymbol{\phi})$ by minimizing the KL-divergence to the predictive posterior of the ensemble:

$$\mathcal{L}^{\mathrm{EnD}}(\boldsymbol{\phi}, \mathcal{D}_{\mathtt{ens}}) = \mathbb{E}_{\hat{\mathrm{p}}(\boldsymbol{x})}\Big[\mathrm{KL}\big[\mathrm{P}(y|\boldsymbol{x}, \mathcal{D}) \,||\, \mathrm{P}(y|\boldsymbol{x}; \boldsymbol{\phi})\big]\Big] \tag{6}$$

This approach has been applied to tasks such as image classification and machine translation [12, 18]. While distillation allows a single model to capture the ensemble's predictive quality and estimates of *total uncertainty* at low cost, information about ensemble diversity is lost. Consequently, we cannot obtain estimates of *knowledge uncertainty*, which are particularly useful for anomaly detection [7, 13].

Recently, a distillation technique called *Ensemble Distribution Distillation* (EnD$^2$) was proposed [13]. Here, both the mean and diversity of an ensemble are captured within a single model via distillation into a Prior Network [19], which is a model that parameterizes the Dirichlet distribution:

$$\mathrm{p}(\boldsymbol{\pi}|\boldsymbol{x}; \boldsymbol{\phi}) = \mathtt{Dir}(\boldsymbol{\pi}|\boldsymbol{\alpha}), \boldsymbol{\alpha} = e^{\boldsymbol{z}}, \boldsymbol{z} = \boldsymbol{f}(\boldsymbol{x}; \boldsymbol{\phi}), \alpha_c > 0, \alpha_0 = \sum_{c=1}^{K}\alpha_c \tag{7}$$

Distribution distillation is done in two steps. Firstly, a *transfer dataset* $\mathcal{D}_{\mathtt{ens}} = \{\boldsymbol{x}^{(i)}, \boldsymbol{\pi}^{(i,1:M)}\}_{i=1}^{N} \sim \hat{\mathrm{p}}(\boldsymbol{x}, \boldsymbol{\pi})$ is composed of the inputs $\boldsymbol{x}^{(i)}$ from the training set $\mathcal{D} = \{\boldsymbol{x}^{(i)}, y^{(i)}\}_{i=1}^{N}$ and the ensemble's output distributions $\{\boldsymbol{\pi}^{(i,1:M)}\}_{i=1}^{N}$ for each input. Secondly, given this transfer set, the model $\mathrm{p}(\boldsymbol{\pi}|\boldsymbol{x}; \boldsymbol{\phi})$ is trained by minimizing the negative log-likelihood of each categorical distribution $\boldsymbol{\pi}^{(im)}$:

$$\mathcal{L}^{\mathrm{EnD}^2}(\boldsymbol{\phi}, \mathcal{D}_{\mathtt{ens}}) = -\mathbb{E}_{\hat{\mathrm{p}}(\boldsymbol{x})}\big[\mathbb{E}_{\hat{\mathrm{p}}(\boldsymbol{\pi}|\boldsymbol{x})}[\ln \mathrm{p}(\boldsymbol{\pi}|\boldsymbol{x}; \boldsymbol{\phi})]\big] \tag{8}$$

Given a distribution-distilled Prior Network, the predictive distribution is given by the expected categorical distribution $\hat{\boldsymbol{\pi}}$ under the Dirichlet prior:

$$\mathrm{P}(y = \omega_c|\boldsymbol{x}^*; \boldsymbol{\phi}) = \mathbb{E}_{\mathrm{p}(\boldsymbol{\pi}|\boldsymbol{x}^*; \boldsymbol{\phi})}[\mathrm{P}(y = \omega_c|\boldsymbol{\pi})] = \pi_c = \frac{\alpha_c}{\sum_{k=1}^{K}\alpha_k} = \frac{e^{z_c}}{\sum_{k=1}^{K}e^{z_k}} \tag{9}$$

Measures of *total* and *knowledge uncertainty* are obtained by considering the mutual information between the prediction $y$ and the parameters of $\boldsymbol{\pi}$ of the categorical:

$$\underbrace{\mathcal{I}[y, \boldsymbol{\pi}|\boldsymbol{x}^*; \boldsymbol{\phi}]}_{\text{Knowledge Uncertainty}} = \underbrace{\mathcal{H}\big[\mathbb{E}_{\mathrm{p}(\boldsymbol{\pi}|\boldsymbol{x}^*; \boldsymbol{\phi})}[\mathrm{P}(y|\boldsymbol{\pi})]\big]}_{\text{Total Uncertainty}} - \underbrace{\mathbb{E}_{\mathrm{p}(\boldsymbol{\pi}|\boldsymbol{x}^*; \boldsymbol{\phi})}\big[\mathcal{H}[\mathrm{P}(y|\boldsymbol{\pi})]\big]}_{\text{Expected Data Uncertainty}} \tag{10}$$

Alternative measures, such as RMI, can also be obtained, as detailed in appendix C.

An alternative approach to capture ensemble diversity is to distill an ensemble into a mixture model which yields a separate output head for each ensemble member [14, 15]. However, the principal downside of this approach is that it attempts to model the ensemble in excessive detail, as it models the behaviour of *each* ensemble member, rather than bulk distributional properties. This requires more flexible and powerful models. As a result, for good performance, it is necessary to split the model into multiple heads at an early stage, which significantly increases the computational and memory complexity. In contrast, EnD$^2$ has a fixed computational and memory cost of one model, regardless of the size of the original ensemble, as it models the ensemble's bulk distributional properties.

# 3 Analysis of first-order gradients

The previous section described how EnD$^2$ can be done by maximising the log-likelihood of the ensemble's outputs under the Dirichlet prior. However, we empirically observed significant convergence issues when optimizing this criterion on tasks with large numbers of classes. As neural networks are predominantly trained using gradient-based optimization, we examine the gradients of the Dirichlet negative log-likelihood loss and propose an alternative training approach.

Our analysis considers the following three situations. First, a Prior Network *initialization* which yields a uniform Dirichlet distribution ($\boldsymbol{\alpha}^{\text{init}} = \mathbf{1}$). The target categorical distribution whose probability is being maximized is a sparse K-length vector of probabilities:

$$\boldsymbol{\pi}^{\text{tgt}} = \left[ 1 - \epsilon, \epsilon/(K-1), \epsilon/(K-1), \cdots \right]^{\text{T}}, \quad \epsilon = 1\text{e-4} \tag{11}$$

Second, a Prior Network *near convergence* with the following output distribution:

$$\boldsymbol{\alpha}^{\text{conv}} = \boldsymbol{\pi}^{\text{conv}} \cdot \alpha_0, \ \alpha_0 = 9\text{e4}, \quad \boldsymbol{\pi}^{\text{conv}} = \left[ 1 - 5\epsilon, \frac{5\epsilon}{K-1}, \frac{5\epsilon}{K-1}, \cdots \right]^{\text{T}} \tag{12}$$

Third, a Prior Network which has made a strong misclassification — a situation which could occur in the middle of training, far from convergence:

$$\boldsymbol{\alpha}^{\text{misc}} = \boldsymbol{\pi}^{\text{misc}} \cdot \alpha_0, \ \alpha_0 = 9\text{e4}, \quad \boldsymbol{\pi}^{\text{misc}} = \left[ \frac{5\epsilon}{K-1}, \frac{5\epsilon}{K-1}, \cdots, 1 - 5\epsilon \right]^{\text{T}} \tag{13}$$

We consider the following training criteria in this analysis: categorical cross entropy[3], Dirichlet negative log-likelihood, Dirichlet Forward and Reverse KL-divergence.

$$\mathcal{L}^{\text{Cat-KL}} = -\sum_{k=1}^{K} \hat{\pi}_k \ln\left(\frac{\alpha_k}{\alpha_0}\right), \quad \mathcal{L}^{\text{Dir-NLL}} = \sum_{k=1}^{K} \Gamma(\alpha_k) - (\alpha_k - 1) \sum_{m=1}^{M} \frac{\ln \pi_k^{(m)}}{M} - \Gamma(\alpha_0)$$

$$\mathcal{L}^{\text{Dir-KL}} = \sum_{k=1}^{K} \Gamma(\alpha_k) - \sum_{k=1}^{K} \Gamma(\beta_k) + \Gamma(\beta_0) - \Gamma(\alpha_0) + \sum_{k=1}^{K} (\beta_k - \alpha_k)\Big(\psi(\beta_k) - \psi(\beta_0)\Big) \tag{14}$$

$$\mathcal{L}^{\text{Dir-RKL}} = \sum_{k=1}^{K} \Gamma(\beta_k) - \sum_{k=1}^{K} \Gamma(\alpha_k) + \Gamma(\alpha_0) - \Gamma(\beta_0) + \sum_{k=1}^{K} (\alpha_k - \beta_k)\Big(\psi(\alpha_k) - \psi(\alpha_0)\Big)$$

The first loss is used for standard distillation [12], which is empirically known to converge well. The second is the loss function originally proposed for EnD$^2$ [13]. The third and fourth loss functions, the Dirichlet forward and reverse KL-divergence, are analyzed as they relate to our proposed solution in the latter part of this section. The *target* Dirichlet distribution for these two loss functions has parameters $\boldsymbol{\beta} = \boldsymbol{\pi}^{\text{tgt}} \cdot \beta_0$ where $\beta_0 = 1\text{e5}$. Full derivations are available in appendix A.

Let's examine the gradients of each criterion presented in equation 14 with respect to the logits.[4] We introduce the ratio $\rho$ of the gradient with respect to the logit $z_1$, the high-probability class of interest, to the average of the absolute values of gradients with respect the logits $z_{2:K}$ of the low-probability tail-classes. The ratio $\rho$ represents the relative contribution of the gradients with respect to the class we are interested in modelling to the long tail.

$$\rho = \frac{K-1}{K} \cdot \left|\frac{\partial \mathcal{L}}{\partial z_1}\right| \Big/ \left(\sum_{k=2}^{K} \left|\frac{\partial \mathcal{L}}{\partial z_k}\right|\right) \tag{15}$$

We normalize by number of classes $K$, as this improves interpretability by making $\rho$ of $\mathcal{L}^{\text{Cat-KL}}$ plateau. This makes it easier to compare to the behaviour of $\rho$ for other criteria.

Figure 1 shows that, at initialization, as the number of classes is increased, the categorical cross-entropy loss primarily focuses on the high probability class and ignores the long tail. In contrast, the Dirichlet NLL displays a diminishing contribution. As the loss is very sensitive to minor perturbations of tail-class probabilities, it will focus on modelling the high-probability classes only after it *perfectly*

---

[3]Which is equivalent to categorical Forward KL-divergence.

[4]We remind the reader that the $\alpha_k = e^{z_k}$, where $z_k$ are the logits.

models the long tail. This means that on complex tasks the model is perpetually stuck modelling the probabilities of tail classes. Note that even near convergence, the ratio $\rho$ is far smaller for $\mathcal{L}^{\text{Dir-NLL}}$ than for categorical cross-entropy. Finally, if a significant error is made, $\rho$ remains large for cross-entropy, but becomes very small for Dirichlet NLL as the number of classes increases. This shows that a desirable property of the loss which ensures good convergence is that the ratio $\rho$ is high and either constant or increasing as the number of classes grows, otherwise the model focuses on modelling the distribution of tail-class probabilities across the ensemble.

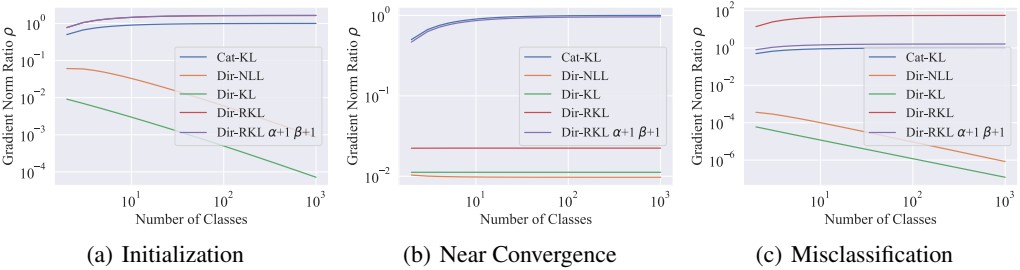

| (a) Initialization | (b) Near Convergence | (c) Misclassification |

Figure 1: Gradient ratio analysis of distillation and distribution distillation criteria.

Figure 1 also displays the ratio $\rho$ for both the forward and reverse Dirichlet KL-divergence losses. Clearly, the forward KL-divergence displays the same issues as the Dirichlet NLL loss and $\rho$ continues to decrease as the number of classes in increased. This is unsurprising, as the negative log-likelihood is equivalent to the KL-divergence in the limit. However, the *reverse KL-divergence* displays the desirable properly that $\rho$ grows and stabilizes as the number of classes is increased in all situations. This suggests that if we were to minimize the *reverse KL-divergence* to an appropriately chosen target Dirichlet distribution, then we could avoid convergence issues.

## 4  Distribution Distillation via Proxy-Dirichlet target distribution

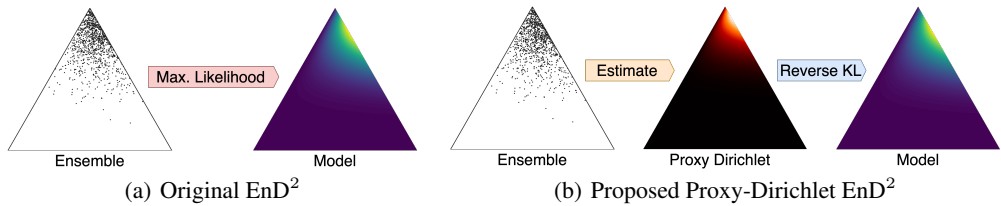

| (a) Original EnD$^2$ | (b) Proposed Proxy-Dirichlet EnD$^2$ |

Figure 2: Schematic of Original and Proposed Distribution Distillation Approaches

The results of the above analysis suggest the reverse KL-divergence criterion has many of the same attributes of the categorical cross-entropy loss — specifically that it focuses on modelling the distribution of probabilities of high-probability classes. Thus, if we could specify an appropriate target *Proxy Dirichlet distribution* which represents the ensemble, then we could accomplish distribution-distillation without optimization difficulties.

To correctly construct a Proxy-Dirichlet target which represents the ensemble, we need to obtain a mean $\hat{\pi}_k$ and precision $\hat{\beta}_0$ derived from the ensemble. While obtaining the ensemble mean is trivial, obtaining an estimate of the precision is more challenging. One approach based on Stirling's approximation is described in [20] and proposes the following estimate:

$$
\hat{\pi}_k(\boldsymbol{x}) = \frac{1}{M}\sum_{m=1}^{M} \text{P}(y = \omega_k | \boldsymbol{x}, \boldsymbol{\theta}^{(m)}), \ \hat{\boldsymbol{\beta}}_k(\boldsymbol{x}) = \frac{\hat{\pi}_k \cdot (K-1)}{\underbrace{2\sum_{k=1}^{K}\hat{\pi}_k(\ln\hat{\pi}_k - \frac{1}{M}\sum_{m=1}^{M}\ln\pi_k^{(m)})}_{\text{Stirling's Approximation}}}
\tag{16}
$$

Thus, the Proxy Dirichlet has both the same mean and *bulk-diversity properties* of the ensemble, such that the measures of diversity derived from the Proxy Dirichlet are similar to those of the original

ensemble. To prevent the Dirichlet distributions of both the target and model from entering highly non-linear regimes impacting the stability of the training process, both the model parameters, $\alpha_k$, and target parameters, $\beta_k$, are constrained to be greater than 1. Figure 1 shows that adding this constraint to the concentration parameters of the Proxy-Dirichlet and the model improves $\rho$, especially near convergence. Altogether, this yields the following criterion:

$$\mathcal{L}^{\text{PD-EnD}^2} = \text{KL}[\text{p}(\boldsymbol{\pi}|\boldsymbol{x}, \boldsymbol{\phi}) \| \text{p}(\boldsymbol{\pi}|\hat{\boldsymbol{\beta}}+\mathbf{1})] \propto \underbrace{\mathbb{E}_{\text{p}(\boldsymbol{\pi}|\boldsymbol{x},\boldsymbol{\phi})}\Big[-\sum_{k=1}^{K}\hat{\pi}_k \ln \pi_k\Big]}_{\text{Reconstruction term}} + \underbrace{\frac{\text{KL}[\text{p}(\boldsymbol{\pi}|\boldsymbol{x},\boldsymbol{\phi})\|\text{p}(\boldsymbol{\pi}|\mathbf{1})]}{\hat{\beta}_0}}_{\text{Prior}} \quad (17)$$

By dividing through by $\hat{\beta}_0$ the above criterion is made to have the same form as an ELBO[5]. This serves two purposes. First, it becomes apparent that the reconstruction term is an upper bound on categorical cross entropy, which explains why they share similar properties. Second, normalizing by $\hat{\beta}_0$ allows this criterion to have the same dynamic range the KL-divergence loss used for standard distillation, and therefore use the same hyperparameters. The proposed approach is evaluated on synthetic data in appendix B.

We want to stress that while this criterion is similar to the one proposed in [10], the underlying reason for its use is entirely different. Here, the issue is due to large gradients from tail classes, while [10] uses a reverse-KL criterion to avoid *inducing in expectation* a multi-modal target Dirichlet distribution.

## 5   Experiments

In this section, we evaluate EnD$^2$ via minimization of the reverse KL-divergence to Proxy Dirichlet target, which we refer to as PD-EnD$^2$. We apply PD-EnD$^2$ to ensembles of convolutional networks trained on the ImageNet [21] dataset, ensembles of VGG-Transformer [22] ASR models trained on LibrSpeech [23] and ensembles of Transformer-big [24] models trained on WMT'17 En-De, which feature 1000, 5000 and 40,000 classes, respectively. Our goal is to demonstrate that given an ensemble, we can successfully distribution-distill it into a single model on tasks with many classes, which was previously impossible. We do not provide results for EnD$^2$ accomplished by optimizing Dirichlet negative log-likelihood or Dirichlet forward KL-divergence because these approaches do not even begin to converge on the tasks considered. Finally, we would like to highlight that providing over-tuned SOTA predictive performance or OOD detection performance on the chosen tasks is not the focus of this research [6].

**Setup** We consider three large-scale tasks involving classification: image classification, speech-to-text transduction and text-to-text transduction. For each task, we first train a Deep Ensemble [4] of regular models and then distribution-distill it with PD-EnD$^2$. We consider the following baselines:

- **Single** refers to the average performance across 10 individual, independently evaluated models.

- **Ensemble** refers to the performance of a Deep Ensemble, which is currently considered to be both the best and the simplest ensemble method [4, 5, 6].

- **EnD** or **SEnD** refer to ensemble distillation via minimizing the KL divergence between the student and the ensemble's mean for image classification [12] and machine translation [18], respectively.

We do not use multi-head approaches [14, 15, 25], for distilling each separate ensemble member, as the computational overhead necessary to individually model each ensemble member with high fidelity on these tasks is significant and it is unclear how to extend them to structured tasks. In all experiments with PD-EnD$^2$, we add 1 both to the predicted parameters of the Dirichlet distribution and the Dirichlet proxy parameters, as discussed in the previous section.

To do distribution-distillation for NMT, which is an autoregressive structured prediction task, we extend Prior Networks and EnD$^2$ to such models and develop *Sequence Ensemble Distribution Distillation* (SEnD$^2$). Models trained with Proxy-Dirichlet targets are referred to as PD-SEnD$^2$.

---

[5]A full derivation is available in appendix A

[6]Code and training configurations are available at `https://github.com/yandex-research/proxy-dirichlet-distillation`.

This extension involves modifications to both the model and derivation of associated sequence-level uncertainty measures [17]. While conceptually straightforward, this modification is extensive and mathematically involved, and therefore instead fully detailed in appendix C.

We evaluate the proposed approach and the baseline in two ways. First, we evaluate their predictive performance and calibration. This assesses how useful the resulting models are to the task at hand. We expect that a successfully distribution-distilled model should have comparable predictive performance to standard. ensemble distillation. Second, we evaluate the measures of uncertainty which the models yield on the the task of out-of-distribution detection. Here we consider a range of *distributionally shifted*, or out-of-distribution, datasets on which we expect the models to either perform poorly, or not to be able to perform at all. The goal is to be able to discriminate between the in-domain and out-of-distribution datasets using measures of uncertainty. We expect that a successfully distribution-distilled model will yield similar performance to the ensemble, especially using measures of *knowledge uncertainty*.

**Large-scale image classification**

We begin by examining PD-EnD$^2$ on the ImageNet [21] dataset, which features 1000 classes. This represents an increase in the scale of the task in in terms of dataset size, image size and, most importantly, number of classes, relative to the tasks considered in the original work on Ensemble Distribution Distillation [13]. Here, we distribution-distill an ensemble of 10 ResNet-50 [26] models. To train each ResNet-50 model, we use the standard training setup outlined in [27, 28]. Specifically, we train for 90 epochs using stochastic gradient descent with a momentum of 0.9 and a learning rate of $0.1 \times B/256$, where B is the per-device batch size multiplied by the number of GPUs. In our experiments, we use a single-GPU batch size of 256 and 8 NVIDIA V100 GPUs. The learning rate is divided by 10 every 30 epochs. For data augmentations, we use a combination of random resized crops and horizontal flips implemented in the Albumentations library [29]. In all experiments, we found it beneficial to initialize the last batch normalization $\gamma$ in each residual branch to zero, which agrees with previous results [28, 30, 31]. Additional results are provided in appendix E.

To evaluate the predictive performance of the proposed method, we measure the classification accuracy and Expected Calibration Error (ECE) on the original ImageNet validation subset [21] and on a range of distributionally shifted datasets. Specifically, we use natural adversarial examples from ImageNet-A [32], corrupted and perturbed versions of the ImageNet validation data from ImageNet-C [33], and artistic renditions from ImageNet-R [34]. These datasets feature the same classes as ImageNet, but they are mismatched in different ways. This assesses how robust the proposed methods are to distributional shift [35], or mismatch, between training and evaluation data. The metrics on ImageNet-C are averaged over all degrees of corruption.

Results in Table 1 show that EnD$^2$ is capable of accurate emulation of the ensemble in terms of classification performance. In terms of accuracy, the method displays results on par or slightly better than standard distillation, and typically better than a single model. The only exception is on the ImageNet-A dataset, where single models achieve best performance. It is likely that the natural adversarial attacks are such that the models in the ensemble become correlated and the error is reinforced. In terms of calibration, the distribution distilled models are closest to the Ensemble. Furthermore, on the in-domain set they outperform the ensemble. These results clearly demonstrate that PD-EnD$^2$ is successfully able to produce models which are comparable to quality to ensemble distillation, but are better calibrated.

Table 1: Prediction quality results for image classification.

| Model | ImageNet-val | | ImageNet-A | | ImageNet-C | | ImageNet-R | |
|---|---|---|---|---|---|---|---|---|
| | Acc | ECE | Acc | ECE | Acc | ECE | Acc | ECE |
| Single | $75.9_{\pm0.1}$ | $4.8_{\pm0.1}$ | $4.4_{\pm0.2}$ | $51.1_{\pm0.3}$ | $39.1_{\pm0.7}$ | $11.3_{\pm0.7}$ | $35.0_{\pm0.2}$ | $21.3_{\pm0.4}$ |
| EnD | $77.0_{\pm0.1}$ | $\mathbf{1.6}_{\pm0.1}$ | $3.7_{\pm0.1}$ | $46.8_{\pm0.2}$ | $40.6_{\pm0.4}$ | $5.8_{\pm0.5}$ | $37.0_{\pm0.3}$ | $15.8_{\pm0.5}$ |
| PD-EnD$^2$ | $77.1_{\pm0.1}$ | $\mathbf{1.6}_{\pm0.1}$ | $3.7_{\pm0.2}$ | $42.9_{\pm0.2}$ | $40.6_{\pm0.5}$ | $4.7_{\pm0.4}$ | $36.9_{\pm0.2}$ | $12.1_{\pm0.5}$ |
| Ensemble | $\mathbf{79.0}_{\pm\text{NA}}$ | $2.3_{\pm\text{NA}}$ | $3.9_{\pm\text{NA}}$ | $42.0_{\pm\text{NA}}$ | $\mathbf{43.5}_{\pm\text{NA}}$ | $4.5_{\pm\text{NA}}$ | $\mathbf{38.8}_{\pm\text{NA}}$ | $\mathbf{9.8}_{\pm\text{NA}}$ |

We evaluate the measures of uncertainty produced by the proposed approach and the baselines on the task of out-of-distribution detection. As discussed above, the goal is to discriminate between in-domain and distributionally shifted (OOD) data using measures of uncertainty. We consider ImageNet

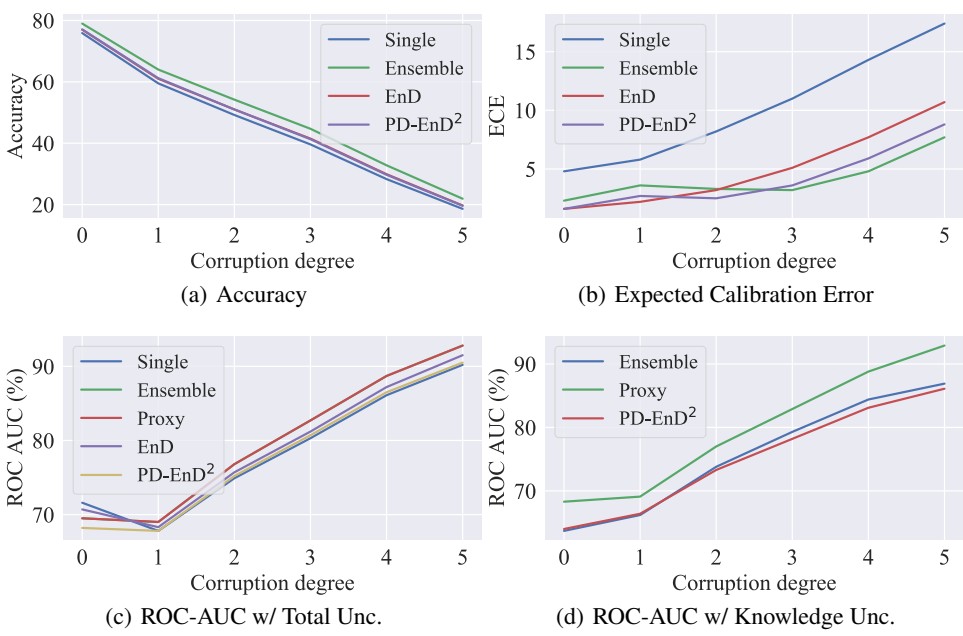

(a) Accuracy

(b) Expected Calibration Error

(c) ROC-AUC w/ Total Unc.

(d) ROC-AUC w/ Knowledge Unc.

Figure 3: Performance of image classification models depending on the level of ImageNet-C corruption. No corruption corresponds to the original ImageNet validation data.

A,C and R as well as ImageNet-O [32], which features altogether new classes not seen in ImageNet, as out-of-distribution data. We use the entropy of the predictive posterior and mutual information (equation 10) as estimates of *total* and *knowledge uncertainty*, respectively. Out-of-distribution detection performance is assessed in terms of Receiver Operating Characteristic area under curve (ROC AUC). We additionally verify how well the Proxy Dirichlet target approximates the ensemble by assessing how well the measures of uncertainty which it yields perform on this task.

Results in Table 2 show several trends. Firstly, the Proxy-Dirichlet distillation can closely match the performance of the ensemble on all datasets except ImageNet-O, which seems to be particularly difficult. Secondly, in all cases the distribution-distilled model is able to obtain performance close to that of the ensemble and generally yields similar behaviour. This clearly demonstrates that PD-EnD$^2$ can successfully train a model to emulate an ensemble.

Table 2: Out-of-distribution detection results for image classification.

|  | ImageNet-O | | ImageNet-A | | ImageNet-C | | ImageNet-R | |
|---|---|---|---|---|---|---|---|---|
|  | Total | Knowledge | Total | Knowledge | Total | Knowledge | Total | Knowledge |
| Single | $50.7_{\pm0.3}$ | - | $85.8_{\pm0.1}$ | - | $79.9_{\pm0.4}$ | - | $83.0_{\pm0.2}$ | - |
| EnD | $48.8_{\pm0.6}$ | - | $87.2_{\pm0.1}$ | - | $80.8_{\pm0.3}$ | - | $83.9_{\pm0.2}$ | - |
| PD-EnD$^2$ | $52.1_{\pm0.2}$ | $53.2_{\pm0.3}$ | $86.8_{\pm0.0}$ | $84.7_{\pm0.1}$ | $80.2_{\pm0.3}$ | $77.4_{\pm0.4}$ | $83.7_{\pm0.1}$ | $81.9_{\pm0.1}$ |
| Ensemble | $\mathbf{54.6}_{\pm\text{NA}}$ | $61.2_{\pm\text{NA}}$ | $\mathbf{88.8}_{\pm\text{NA}}$ | $87.2_{\pm\text{NA}}$ | $\mathbf{82.0}_{\pm\text{NA}}$ | $78.1_{\pm\text{NA}}$ | $\mathbf{86.1}_{\pm\text{NA}}$ | $84.4_{\pm\text{NA}}$ |
| Proxy | $\mathbf{54.6}_{\pm\text{NA}}$ | $56.2_{\pm\text{NA}}$ | $\mathbf{89.0}_{\pm\text{NA}}$ | $86.5_{\pm\text{NA}}$ | $\mathbf{82.0}_{\pm\text{NA}}$ | $81.1_{\pm\text{NA}}$ | $\mathbf{86.1}_{\pm\text{NA}}$ | $86.3_{\pm\text{NA}}$ |

In addition to the tables above, we provide Figure 3, which shows a detailed breakdown by corruption strength of the predictive and OOD detection performance on ImageNet-C. Results show that as the degree of corruption increases, all models suffer a drop in accuracy and calibration quality; notably, EnD and PD-EnD$^2$ have the same calibration performance on original data, but PD-EnD$^2$ has lower calibration errors for the highest degrees of corruption. Unsurprisingly, the greater the distributional shift from the original training images, the better the models are at out-of-distribution detection. Overall, the distribution-distilled model clearly follows the behaviour of the ensemble in all metrics.

**Speech Recognition**

Table 3: Predictive performance results for speech recognition in terms of WER and NLL.

| Model | LTC | | LTO (ROC-AUC %) | | | | AMI (ROC-AUC %) | | | | C-FR (ROC-AUC %) | | | |
|---|---|---|---|---|---|---|---|---|---|---|---|---|---|---|
| | | | Total | Knowledge | | | Total | Knowledge | | | Total | Knowledge | | |
| | WER | NLL | $\mathcal{H}$ | $\mathcal{M}$ | $\mathcal{I}$ | $\mathcal{K}$ | $\mathcal{H}$ | $\mathcal{M}$ | $\mathcal{I}$ | $\mathcal{K}$ | $\mathcal{H}$ | $\mathcal{M}$ | $\mathcal{I}$ | $\mathcal{K}$ |
| Single | 6.1 | 0.35 | 72.8 | - | - | - | 94.4 | - | - | - | 99.8 | - | - | - |
| SEnD | 6.2 | 0.36 | 72.8 | - | - | - | 90.2 | - | - | - | 99.8 | - | - | - |
| PD-SEnD$^2$ | 7.1 | 0.25 | 76.6 | 75.5 | 50.0 | 75.7 | 94.4 | 75.2 | 50.0 | 76.8 | **99.9** | 98.8 | 50.0 | 98.9 |
| Ensemble | **4.2** | **0.20** | 76.9 | 77.0 | 76.6 | 77.0 | **96.5** | 94.8 | 94.9 | 94.9 | **99.9** | **99.9** | 99.9 | 99.9 |

Now let's consider applying PD-EnD$^2$ to speech recognition. We distill an ensemble of 6 VGG-Transformer models [22] trained on the English speech recordings from LibriSpeech [23]. Autoregressive speech recognition systems are challenging to train even in a standard setting — the VGG-Transformer architecture tends to be unstable in training. Furthermore, speech models tend to yield extremely sharp, over-confident predictions, which both exacerbates the issues with tail-class gradients dominating the gradients from high-probability classes (when training with naive EnD$^2$), and also causes numerical issues when distilling using a Proxy Dirichlet. This makes ASR an interesting "stress-test" task for assessing the proposed solution for Ensemble Distribution Distillation.

We base our training setup on the description given in [22], training for 80 epochs on 8 GPUs with a batch size of 8,000 tokens with the AdaDelta [36] algorithm and a constant learning rate. The dataset is processed with a vocabulary of 5000 sentence-piece tokens. Checkpoints over the last 30 epochs are averaged together. The only significant difference is that we train SEnD$^2$ with the Adam [37] optimizer, using the same learning rate schedule as in the machine translation experiments.

We use four datasets for evaluation: LibriSpeech test-clean and test-other, AMI meeting transcription dataset [38], and French speech recordings from Common Voice [39]. This setup is identical to the one considered in [17]. LibriSpeech test-other represents a noisy dataset, while AMI and Common-Voice French represent significant distributional shifts.

First, we compare the methods in terms of their predictive performance using the Word Error Rate (WER)[7] and test-set negative log-likelihood on the LibriSpeech test-clean datset (LTC). Results in Table 3 show that both distillation and distribution distillation are challenging for ASR: neither model is able to outperform a single model in terms of WER. However, in terms of test-set NLL, the distribution-distilled model (PD-SEnD$^2$) is significantly better (lower NLL) than both a single model and the standard distilled model. Coupled with the fact that the WER is worst for PD-SEnD$^2$, this suggest that the model is underfit, and a different training setup, with longer training and less regularization, should be considered. Nevertheless, these preliminary results show that comparable performance is attainable. We remind the reader that Naive EnD$^2$ does not even begin to converge on the tasks which we consider here. Secondly, we assess the uncertainty estimates via OOD detection. As the in-domain set, we use Librispeech test-clean, and as the OOD sets, we consider LibriSpeech test-other (LTO), AMI and CV French. For out-of distribution detection, we use the same measures of total and knowledge uncertainty as in Section 5, which are detailed in appendix C. Results in table 3 show that the distribution-distilled model is able to yield measures of *total uncertainty* which outperform both a single model and a distilled model. However, measures of knowledge uncertainty do not match the ensemble well, which suggests that the model is either misfit or underfit. In the light of the predictive performance results, the latter is more likely.

**Machine translation**

Now we evaluate PD-EnD on the WMT'17 English-German machine translation dataset with a vocabulary of 40,000 Byte-Pair Encoding tokens [40]. This represents a more challenging task of greater scale than ImageNet. For distillation and distribution distillation we use an ensemble of 10 Transformer-big [41] trained with the setup described in [42]. Specifically, ensemble members are trained for 193,000 steps with Adam [37] on 8 NVIDIA V100 GPUs with a batch size of 4096 tokens per GPU. For both distillation and distribution distillation, we train all models for 20,000 steps with a large batch size of 32K tokens. As our implementation requires fitting all 10 ensemble members in GPU memory, we reduce the immediate batch size for each step to 1024, but compensate for it with gradient accumulation over 32 steps. We use beam search with the beam size equal to 5 for inference.

---

[7]Computed using NIST SCLITE tool.

Table 4: BLEU and out-of-distribution detection results for machine translation.

| Model | BLEU | NLL | NLL Ens-Pred | Permuted | | Speech | | German | | French | |
|---|---|---|---|---|---|---|---|---|---|---|---|
| | | | | Total | Know. | Total | Know. | Total | Know. | Total | Know. |
| Single | $28.8_{\pm0.1}$ | 1.46 | 0.42 | $80.7_{\pm1.5}$ | - | $73.7_{\pm1.2}$ | - | $32.8_{\pm2.8}$ | - | $27.1_{\pm6.3}$ | - |
| Ensemble | $\mathbf{30.1}_{\pm\text{NA}}$ | 1.33 | 0.39 | $83.7_{\pm\text{NA}}$ | $\mathbf{97.4}_{\pm\text{NA}}$ | $67.8_{\pm\text{NA}}$ | $73.7_{\pm\text{NA}}$ | $39.5_{\pm\text{NA}}$ | $\mathbf{82.4}_{\pm\text{NA}}$ | $25.0_{\pm\text{NA}}$ | $\mathbf{73.6}_{\pm\text{NA}}$ |
| SEnD | $29.4_{\pm0.1}$ | 1.52 | 0.4 | $79.5_{\pm1.1}$ | - | $75.9_{\pm0.6}$ | - | $35.4_{\pm1.6}$ | - | $15.6_{\pm3.2}$ | - |
| PD-SEnD$^2$ | $29.5_{\pm0.1}$ | 1.52 | 0.4 | $78.3_{\pm1.6}$ | $97.1_{\pm0.3}$ | $77.0_{\pm0.3}$ | $\mathbf{78.5}_{\pm0.2}$ | $38.3_{\pm1.6}$ | $70.9_{\pm0.7}$ | $15.9_{\pm3.0}$ | $60.1_{\pm3.6}$ |

First, we assess the proposed approach in terms of translation quality on the newstest14 English-German test set, for which we use the BLEU score [43] computed with SacreBLEU [44]. Results in Table 4 confirm that PD-EnD$^2$ is capable of matching regular distillation in terms of translation quality. This shows that the proposed training approach does not suffer from any optimization issues and yields fully converged models of the same quality as standard ensemble distillation.

Second, we assess the calibration performance in terms of NLL both on the reference transcriptions and on the ensemble's transcription (ENS-Pred) of test data. The results show that the ensemble has better (lower) NLL than a single model on reference transcriptions, but both distilled models have worse (higher) NLL than both the ensemble and single model on reference transcriptions. On ensemble transcriptions, both distilled models have better NLLs than the single model. This is expected, as these models were never trained on the reference data — they were always trained on transcriptions provided by the ensemble. Furthermore, note that the NLLs of all models are better (lower) on ensemble predictions than on the references. This is consistent with the observations made in [45] that reference transcriptions are more complex and multimodal, more surprising, therefore would have higher NLLs, while predictions of NMT models are more consistent in style and therefore simpler — they are less 'surprising', and therefore have lower NLL. These results show that while all distilled models consider the references to be more surprising, as their decoders were never exposed to the reference text, the distilled models are more closely matched to the ensemble, and thus yield better performance.

Third, we assess the quality of the uncertainty estimates which our distribution-distilled model yields. For out-of-distribution detection, we use *sequence-level* entropy and reverse mutual information as measures of total and knowledge uncertainty, which have been shown to perform well on this task [17].[8] The in-domain test set is newstest'14 and OOD datasets are as follows: 'permuted' — sentences with permuted tokens in the input, 'speech' — LibriSpeech [23] test-clean English speech transcriptions; 'German' and 'French' — source sentences from newstest'14 in German and French languages respectively. This is the same setup and datasets as were considered by [17]. Results of both distillation methods are averaged over 5 random seeds.

Table 4 clearly shows that in all cases, the PD-SEnD$^2$ model is able to yield measures of uncertainty which outperform SEnD and single models and are highly competitive with the original ensemble. However, the distribution-distilled model doesn't follow the behaviour of the ensemble as closely, which is not surprising, as the task is far more complex than image classification and the output space is exponentially vast.

## 6 Conclusion

This work has examined applying Ensemble Distribution Distillation (EnD$^2$) to large-scale tasks where the number of classes is very high. Training these models is shown to be challenging with often poor convergence properties. We provided an analysis which shows that the standard training criterion, the Dirichlet log-likelihood yields gradients associated with low probability classes that dominate those of the higher probability classes. Thus, during training the model will focus on the distribution of these ensemble tail-class probabilities. To address this problem, we proposed a new training objective which minimizes the reverse KL-divergence to a *Proxy-Dirichlet* target derived from the ensemble. This new criterion resolves the gradient issues of EnD$^2$, which we demonstrate empirically on the ImageNet, LibriSpeech and WMT17 En-De datasets, which have 1000, 5000 and 40,000 classes, respectively. Thus, this work enables Ensemble Distribution Distillation to be applied to tasks with arbitrary numbers of classes and complexity, dramatically broadening the range of applications that can benefit from rapid inference and efficient uncertainty estimation.

---

[8]Further details and derivations are available in appendix C.

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
