# A  Loss Derivation

In this section we provide a more detailed derivation of the proposed loss function (Equation 17).

$$
\begin{aligned}
\mathcal{L}^{\text{PD-EnD}^2} &= \text{KL}[\text{p}(\boldsymbol{\pi}|\boldsymbol{x},\boldsymbol{\phi})\|\text{p}(\boldsymbol{\pi}|\hat{\boldsymbol{\beta}}+\mathbf{1})] \\
&= -\,\mathbb{E}_{\text{p}(\boldsymbol{\pi}|\boldsymbol{x},\boldsymbol{\phi})}\big[\ln \text{p}(\boldsymbol{\pi}|\hat{\boldsymbol{\beta}}+\mathbf{1})\big] + \mathbb{E}_{\text{p}(\boldsymbol{\pi}|\boldsymbol{x},\boldsymbol{\phi})}\big[\ln \text{p}(\boldsymbol{\pi}|\boldsymbol{x},\boldsymbol{\phi})\big] \\
&= -\,\mathbb{E}_{\text{p}(\boldsymbol{\pi}|\boldsymbol{x},\boldsymbol{\phi})}\Big[\sum_{k=1}^{K}\hat{\beta}_k \ln \pi_k\Big] + \text{KL}[\text{p}(\boldsymbol{\pi}|\boldsymbol{x},\boldsymbol{\phi})\|\text{p}(\boldsymbol{\pi}|\mathbf{1})] + Z \\
&= \underbrace{-\mathbb{E}_{\text{p}(\boldsymbol{\pi}|\boldsymbol{x},\boldsymbol{\phi})}\Big[\sum_{k=1}^{K}\hat{\pi}_k \ln \pi_k\Big]}_{\text{Reconstruction term}} + \underbrace{\frac{\text{KL}[\text{p}(\boldsymbol{\pi}|\boldsymbol{x},\boldsymbol{\phi})\|\text{p}(\boldsymbol{\pi}|\mathbf{1})]}{\hat{\beta}_0}}_{\text{Prior}}
\end{aligned}
\tag{18}
$$

We make use of the fact that the negative entropy of the Dirichlet distribution is equivalent to the reverse KL-divergence to a flat Dirichlet, up to an additive constant which doesn't depend on the model. Additionally, we can see that by adding +1 to the target concentration parameters $\hat{\boldsymbol{\beta}}$, we are now minimizing an upper bound to the KL-divergence between the mean and the ensemble. Then we divide through by $\hat{\beta}_0$ and drop the additive constant. This yields a loss which is remarkable similar to an ELBO. Note that the parameter $\hat{\beta}_0$ weights the reconstruction term, which forces the model to produce a sharper Dirichlet [7], and the KL-divergence to a flat Dirichlet, which balances the effect of the reconstruction term.

# B  Synthetic Experiments

In addition to the first-order gradient analysis conducted in section 3, we ran an experiment on a the synthetic spiral dataset used in the original Ensemble Distribution Distillation paper [13]. We considered data with 3, 6, 10 and 20 classes to demonstrate at which point standard $\text{EnD}^2$ begins degradation. Identical hyper-parameters and models were used for both $\text{EnD}^2$ and $\text{PD-EnD}^2$. The spiral dataset is such that the variation in ensemble precision across different input locations can be enormous: rare points right on the arms of the spiral can have several orders of magnitude larger precisions than in other locations. This leads to numerical instabilities and NaN occasionally in $\text{PD-EnD}^2$ due to division by precision. We resolved this by using a single LayerNorm layer just before the final output layer. Using LayerNorm (or similar) for standard $\text{EnD}^2$ via Dirichlet NLL prevented convergence altogether and was not used. We suspect that a more numerically stable implementation of the loss would not require LayerNorm. Note that adding LayerNorm did not improve the results for $\text{PD-EnD}^2$: it only prevented any training run from aborting due to NaNs in the loss.

The results in Table 5 show that on 3 classes both $\text{EnD}^2$ and $\text{PD-EnD}^2$ can match both the ensemble and the single model in terms of error. As the number of classes is increased, the $\text{PD-EnD}^2$ consistently yields lower error rates. In contrast, standard $\text{EnD}^2$ displays growing error rates and failure of convergence for 20 classes. Note, we explicitly did not use the heuristic temperature tricks from [13] in order to compare pure forms of the Dirichlet Likelihood of $\text{PD-EnD}^2$ reverse KL objectives. The results generally agree with the analysis presented in Section 3, where it is clear that even for 10 classes for reverse KL loss $\rho$ is larger than for the Dirichlet-NLL loss.

Table 5: Error rates (%) on heldout data for synthetic data with 3–20 classes

| Model | 3 | 6 | 10 | 20 |
|---|---|---|---|---|
| Single | 1.48 | 1.93 | 0.99 | 0.75 |
| $\text{EnD}^2$ | 1.98 | 2.38 | 4.92 | - |
| $\text{PD-EnD}^2$ | 1.22 | 1.57 | 0.61 | 0.52 |
| Ensemble | 1.1 | 1.6 | 0.5 | 0.5 |

Additionally, we examined the models' median precisions ($\alpha_0$ and compared to the ensemble's approximate precision obtained via Stirling's approximation. We used median over the heldout

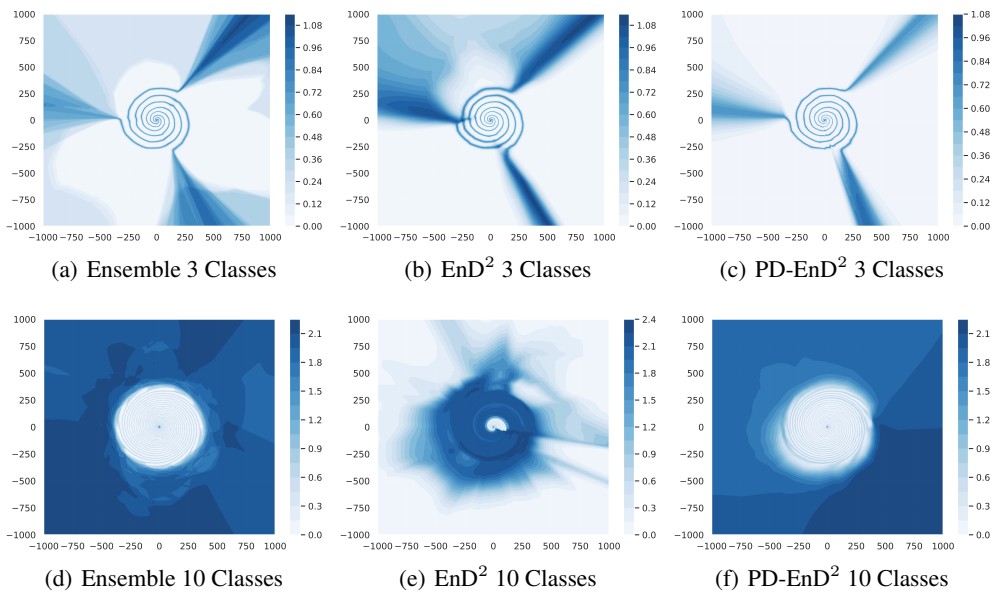

Figure 4: Total Uncertainty heat maps for models with 3 and 10 classes for Ensemble, $EnD^2$ and $PD\text{-}EnD^2$ models.

dataset as the presence of rare inputs points right at the center of the spiral arms, whose precisions were orders of magnitude larger, greatly affected the mean. The results show that standard $EnD^2$ yields low precisions which decrease as the number of classes grows, while $PD\text{-}EnD^2$ consistently yields precisions comparable to the ensemble.

Table 6: Precisions on heldout data for synthetic data with 3-20 classes

| Model | 3 | 6 | 10 | 20 |
|---|---|---|---|---|
| $EnD^2$ | 278.7 | 13.6 | 11.75 | - |
| $PD\text{-}EnD^2$ | 4413.5 | 2791.2 | 2691.5 | 3985.2 |
| Ensemble | 14350.7 | 10275.9 | 8859.8 | 6049.4 |

Finally, we examined the behaviour of the uncertainty estimates produced by the ensemble and the distribution-distilled models using heat maps of total and knowledge uncertainty values across a fine grid of inputs. These results are presented in Figures 4 and 5, which depict the total and knowledge uncertainty, respectively, for the original ensemble and models distribution distilled via $EnD^2$ and $PD\text{-}EnD^2$. Clearly all models behave similarly where there are 3 classes, but $EnD^2$ yields poor behaviour and doesn't match the ensemble for 10 classes. These results generally agree with the plots of $\rho$ versus the number of classes, which show that for standard $EnD^2$ (Dirichet NLL) there is an immediate drop in $\rho$, while for the proposed approach $\rho$ remains high.

## C   Sequence Ensemble Distribution Distillation

In this section we describe how to generalize distribution distillation to autoregressive models, yielding Sequence Ensemble Distributions Distillation ($SEnD^2$). Much of this section is a direct adaption of [17] to Prior Network models.

**Autoregressive Prior Networks** Unlike ensembles of classification models, which can be interpreted as samples from an implicit conditional distribution over discrete output distributions, ensembles of autoregressive models can be interpreted as samples of prefix trees sampled from a distribution over prefix trees conditioned on the input $x$. Unfortunately, it is not possible to explicitly parametrize such a distribution in practice. However, it was shown in [17], ensemble-combination for

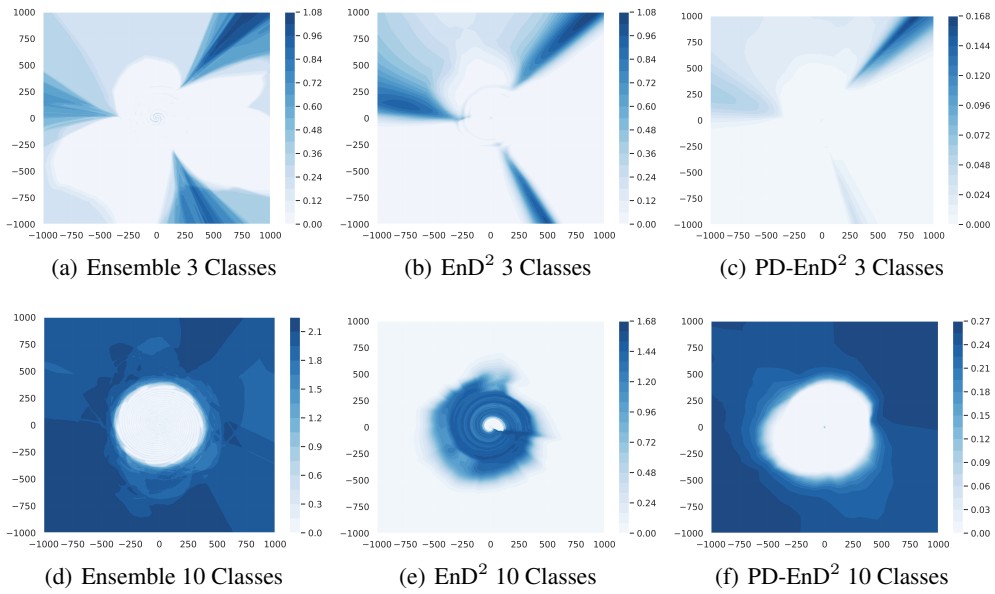

(a) Ensemble 3 Classes      (b) EnD$^2$ 3 Classes      (c) PD-EnD$^2$ 3 Classes

(d) Ensemble 10 Classes      (e) EnD$^2$ 10 Classes      (f) PD-EnD$^2$ 10 Classes

Figure 5: Knowledge Uncertainty heat maps for models with 3 and 10 classes for Ensemble, EnD$^2$ and PD-EnD$^2$ models.

autoregressive models should be done as a *token-level* Bayesian model average (BMA). Ensembles of auto-regressive models can therefore be interpreted as samples from a distribution over output distributions conditioned on the input and the context.

$$\big\{P(y_l|\boldsymbol{y}_{<l}, \boldsymbol{x}; \boldsymbol{\theta}^{(m)})\big\}_{m=1}^{M} \to p(\boldsymbol{\pi}_l|\boldsymbol{y}_{<l}, \boldsymbol{x}; \boldsymbol{\phi}) \tag{19}$$

Let's examine how to emulate an ensemble of auto-regressive models using Prior Networks. Consider an *autoregressive* Prior Network (APN) $p(\boldsymbol{\pi}_l|\boldsymbol{y}_{<l}, \boldsymbol{x}; \boldsymbol{\phi})$ which yields a distribution over output distributions $\boldsymbol{\pi}_l = \big[P(y_l = \omega_1), \cdots, P(y_l = \omega_K)\big]^{\mathrm{T}}$ at every timestep:

$$p(\boldsymbol{\pi}_l|\boldsymbol{y}_{<l}, \boldsymbol{x}; \boldsymbol{\phi}) = \mathtt{Dir}(\boldsymbol{\pi}|\boldsymbol{\alpha}^{(l)}), \quad \boldsymbol{\alpha}^{(l)} = \boldsymbol{f}(\boldsymbol{y}_{<l}, \boldsymbol{x}; \boldsymbol{\phi}), \quad \alpha_c^{(l)} > 0, \ \alpha_0^{(l)} = \sum_{c=1}^{K} \alpha_c^{(l)} \tag{20}$$

Given this model, the Token-level BMA can be straightforwardly emulated as follows:

$$P(\boldsymbol{y}|\boldsymbol{x}, \mathcal{D}) = \prod_{l=1}^{L} \mathbb{E}_{\mathtt{q}(\boldsymbol{\theta})}\Big[P(y_l|\boldsymbol{y}_{<l}, \boldsymbol{x}, \boldsymbol{\theta})\Big] \approx \prod_{l=1}^{L} \mathbb{E}_{p(\boldsymbol{\pi}_l|\boldsymbol{y}_{<l}, \boldsymbol{x}; \boldsymbol{\phi})}[P(y_l|\boldsymbol{\pi}_l)] = P(\boldsymbol{y}|\boldsymbol{x}; \boldsymbol{\phi}) \tag{21}$$

We obtain predictions by running *beam search decoding* with a beam of size $S$.

**Measures of Uncertainty** Let's examine how given this model we can obtain measures of sequence-level *total* and *knowledge* uncertainty. *Total Uncertainty* will be given by the sequence-level entropy:

$$\mathcal{H}\big[P(\boldsymbol{y}|\boldsymbol{x}; \boldsymbol{\phi})\big] = \frac{1}{L}\mathbb{E}_{P(\boldsymbol{y}|\boldsymbol{x}; \boldsymbol{\phi})}\big[P(\boldsymbol{y}|\boldsymbol{x}; \boldsymbol{\phi})\big] = \frac{1}{L}\sum_{l=1}^{L}\mathbb{E}_{P(\boldsymbol{y}_{<l}|\boldsymbol{x}; \boldsymbol{\phi})}\big[\mathcal{H}\big[P(y_l|\boldsymbol{y}_{<l}, \boldsymbol{x}; \boldsymbol{\phi})\big]\big] \tag{22}$$

Here, we use the chain-rule of entropy to decompose the sequence level entropy into a sum of expected token-level entropies. For an Autoregressive Prior Network, we will estimate this as follows:

$$\begin{aligned}
\hat{\mathcal{H}}^{(S)}\big[P(\boldsymbol{y}|\boldsymbol{x}; \boldsymbol{\phi})\big] &\approx \frac{1}{SL}\sum_{s=1}^{S}\sum_{l=1}^{L}\hat{\mathcal{H}}\big[P(y_l|\boldsymbol{y}_{<l}, \boldsymbol{x}; \boldsymbol{\phi})\big], \quad \boldsymbol{y}_{<l}^{(s)} \in \boldsymbol{y}^{(s)} \sim P(\boldsymbol{y}|\boldsymbol{x}; \boldsymbol{\phi}) \\
&= \frac{1}{SL}\sum_{s=1}^{S}\sum_{l=1}^{L}\Big[-\sum_{k=1}^{K}\frac{\hat{\alpha}_k^{(sl)}}{\hat{\alpha}_0^{(sl)}}\ln\frac{\hat{\alpha}_k^{(sl)}}{\hat{\alpha}_0^{(sl)}}\Big]
\end{aligned} \tag{23}$$

Here we combine information from multiple beam search hypotheses by averaging across all hypotheses in the beam, as done in [17].

To obtain estimates of *knowledge uncertainty*, we will consider sequence level measures of ensemble diversity [17]. We will consider mutual information $\mathcal{I}$, expected pairwise KL-divergence $\mathcal{K}$ and reverse mutual information $\mathcal{M}$. First, we will express sequence-level mutual information as a sum of token-level mutual information, averaged across all hypotheses within a beam:

$$
\begin{aligned}
\hat{\mathcal{I}}^{(S)}[\boldsymbol{y}, \boldsymbol{\pi}_{1:L}|\boldsymbol{x}] &\approx \frac{1}{SL} \sum_{s=1}^{S} \sum_{l=1}^{L} \hat{\mathcal{I}}[y_l, \boldsymbol{\pi}_l|\boldsymbol{y}_{<l}, \boldsymbol{x}], \quad \boldsymbol{y}_{<l}^{(s)} \in \boldsymbol{y}^{(s)} \sim \mathtt{P}(\boldsymbol{y}|\boldsymbol{x}; \hat{\boldsymbol{\phi}}) \\
&= \frac{1}{SL} \sum_{s=1}^{S} \sum_{l=1}^{L} \Big[ -\sum_{k=1}^{K} \frac{\hat{\alpha}_k^{(sl)}}{\hat{\alpha}_0^{(sl)}} \ln \frac{\hat{\alpha}_k^{(sl)}}{\hat{\alpha}_0^{(sl)}} + \sum_{k=1}^{K} \frac{\hat{\alpha}_k^{(sl)}}{\hat{\alpha}_0^{(sl)}} \big(\psi(\hat{\alpha}_k^{(sl)}+1) - \psi(\hat{\alpha}_0^{(sl)}+1)\big) \Big]
\end{aligned}
\tag{24}
$$

Note, that this is an *asymptotically inexact approximation*, as it is intractable to compute a true estimate of sequence-level mutual information, as discussed in [17]. We can obtain an approximation for sequence level EPKL with similar properties:

$$
\begin{aligned}
\hat{\mathcal{K}}^{(S)}[\boldsymbol{y}, \boldsymbol{\pi}_{1:L}|\boldsymbol{x}] &\approx \frac{1}{SL} \sum_{s=1}^{S} \sum_{l=1}^{L} \hat{\mathcal{K}}[y_l, \boldsymbol{\pi}_l|\boldsymbol{y}_{<l}^{(s)}, \boldsymbol{x}], \quad \boldsymbol{y}_{<l}^{(s)} \in \boldsymbol{y}^{(s)} \sim \mathtt{P}(\boldsymbol{y}|\boldsymbol{x}; \hat{\boldsymbol{\phi}}) \\
&= \frac{1}{SL} \sum_{s=1}^{S} \sum_{l=1}^{L} \frac{K-1}{\hat{\alpha}_0^{(sl)}}
\end{aligned}
\tag{25}
$$

Finally, we can compute sequence level *reverse mutual information*, which we express as the sum of token-level revese mutual information. It is important to highlight that for this measure we obtain an *asymptotically exact approximation* which is partciularly useful for strucutred tasks [17]:

$$
\begin{aligned}
\hat{\mathcal{M}}^{(S)}[\boldsymbol{y}, \boldsymbol{\pi}_{1:L}|\boldsymbol{x}] &\approx \frac{1}{SL} \sum_{s=1}^{S} \sum_{l=1}^{L} \hat{\mathcal{M}}[y_l, \boldsymbol{\pi}_l|\boldsymbol{y}_{<l}, \boldsymbol{x}], \quad \boldsymbol{y}_{<l}^{(s)} \in \boldsymbol{y}^{(s)} \sim \mathtt{P}(\boldsymbol{y}|\boldsymbol{x}; \hat{\boldsymbol{\phi}}) \\
&= \frac{1}{SL} \sum_{s=1}^{S} \sum_{l=1}^{L} \Big[ -\sum_{k=1}^{K} \frac{\hat{\alpha}_k^{(sl)}}{\hat{\alpha}_0^{(sl)}} \big(\psi(\hat{\alpha}_k^{(sl)}) - \psi(\hat{\alpha}_0^{(sl)})\big) + \sum_{k=1}^{K} \frac{\hat{\alpha}_k^{(sl)}}{\hat{\alpha}_0^{(sl)}} \ln \frac{\hat{\alpha}_k^{(sl)}}{\hat{\alpha}_0^{(sl)}} \Big]
\end{aligned}
\tag{26}
$$

**PD-SEnD$^2$ training criterion** We can extend the proposed Proxy-Dirichlet reverse-KL-divergence training criterion to sequence distribution distillation by constructing a Proxy Dirichlet target with parameters $\hat{\boldsymbol{\pi}}_l$ and $\beta_0^{(l)}$ for each time-step $l$. Then, the loss is trivially an expectation over all time-steps:

$$
\mathcal{L}^{\text{PD-SEnD}^2} = \frac{1}{L} \sum_{l=1}^{L} \Big[ -\mathbb{E}_{\mathtt{p}(\boldsymbol{\pi}_l|\boldsymbol{y}_{<l}, \boldsymbol{x}, \boldsymbol{\phi})} \Big[ \sum_{k=1}^{K} \hat{\pi}_k^{(l)} \ln \pi_k^{(l)} \Big] + \frac{\mathtt{KL}[\mathtt{p}(\boldsymbol{\pi}_l|\boldsymbol{y}_{<l}, \boldsymbol{x}, \boldsymbol{\phi}) \| \mathtt{p}(\boldsymbol{\pi}_l|\mathbf{1})]}{\hat{\beta}_0^{(l)}} \Big]
\tag{27}
$$

## D  Ensemble Distribution Distillation for Speech Recognition

Table 7 presents the predictive performance and calibration, in terms of NLL, of ASR models on two additional datasets: LibriSpeech test-other and the AMI meeting transcription corpus [38]. The results show that PD-SEnD$^2$ doesn't quite reach the same performance as Sequence Distillation or the ensemble, but isn't far behind and also has the second best NLL performance consistently across all datasets. These results suggest that the model is underfit and alternative training setups should be considered, which we leave to be explored in future work.

## E  Technical Details

To increase the training stability of our method, in all experiments we convert the Dirichlet distribution parameters to a higher numerical precision of 64 bits before computing the loss. This conversion allows the model to express a higher range of possible values, which is often necessary for accurate

Table 7: Predictive performance results for speech recognition in terms of WER and NLL.

| Model | LTC | | LTO | | AMI | |
|---|---|---|---|---|---|---|
| | WER | NLL | WER | NLL | WER | NLL |
| Single | 6.1 | 0.35 | 15.2 | 0.86 | 56.8 | 5.71 |
| Ensemble | **4.2** | **0.20** | **11.3** | **0.48** | **50.4** | 4.05 |
| SEnD | 6.2 | 0.36 | 15.5 | 0.90 | 56.0 | 6.06 |
| PD-SEnD$^2$ | 7.1 | 0.25 | 17.0 | 0.68 | 54.8 | **3.85** |

Table 8: Prediction and calibration results for image classification. MIMO with $M = 2$ was used.

| | ImageNet-val | | | | | ImageNet-A | | | | | ImageNet-C | | | | | ImageNet-R | | | | |
|---|---|---|---|---|---|---|---|---|---|---|---|---|---|---|---|---|---|---|---|---|
| | ACC | NLL | ECE | MCE | Brier | ACC | NLL | ECE | MCE | Brier | ACC | NLL | ECE | MCE | Brier | ACC | NLL | ECE | MCE | Brier |
| Single | $75.9_{\pm0.1}$ | 0.96 | 4.7 | 9.1 | 0.34 | $4.4_{\pm0.2}$ | 6.49 | 51.2 | 90.9 | 1.34 | $39.1_{\pm0.7}$ | 3.43 | 12.0 | 24.8 | 0.76 | $35.0_{\pm0.2}$ | 4.02 | 20.9 | 34.4 | 0.84 |
| MIMO | $76.8_{\pm\text{NA}}$ | 0.91 | 2.7 | 4.7 | 0.32 | $5.2_{\pm\text{NA}}$ | 5.90 | 47.4 | 90.0 | 1.29 | $41.3_{\pm0.1}$ | 3.05 | 8.1 | 18.2 | 0.72 | $37.0_{\pm0.3}$ | 3.67 | 17.6 | 27.7 | 0.79 |
| EnD | $77.0_{\pm0.1}$ | 0.89 | 1.6 | 3.8 | 0.32 | $3.9_{\pm0.1}$ | 5.90 | 46.6 | 90.9 | 1.29 | $40.6_{\pm0.4}$ | 3.09 | 5.8 | 14.3 | 0.72 | $36.8_{\pm0.2}$ | 3.66 | 16.1 | 26.2 | 0.79 |
| EnD$^2$ | $77.1_{\pm0.1}$ | 0.89 | 1.6 | 4.5 | 0.32 | $4.0_{\pm0.2}$ | 5.50 | 42.8 | 90.2 | 1.25 | $40.6_{\pm0.5}$ | 3.11 | 4.6 | 12.3 | 0.72 | $36.9_{\pm0.2}$ | 3.39 | 11.7 | 20.8 | 0.77 |
| Ensemble | $\mathbf{79.0}_{\pm\text{NA}}$ | 0.81 | 2.4 | 6.7 | 0.30 | $3.9_{\pm\text{NA}}$ | 5.59 | 42.0 | 88.9 | 1.23 | $\mathbf{43.5}_{\pm\text{NA}}$ | 2.90 | 4.6 | 9.7 | 0.69 | $\mathbf{38.8}_{\pm\text{NA}}$ | 3.48 | 9.7 | 15.0 | 0.75 |

replication of the ensemble's behavior. In addition, for increased numerical stability we also clip all $\alpha_k$, limiting their maximal value to $\alpha_{\max}/K$, where $\alpha_{\max}$ is the highest possible value of the data type used for parameters. This procedure ensures that the value $\alpha_0$ does not overflow during training. The issue of parameter overflow arises rather sporadically at early stages of training due to the stochasticity of the optimization process: the model predicts high $\alpha_k$ for one class of one example in the batch, while all other outputs have significantly lower magnitudes.

Also, in practice we do not generate the transfer dataset due to the large output space size and the number of training samples, making it quite hard to keep such a dataset on a regular storage device. Instead, we load the ensemble into memory during training and obtain output distributions of its members for each training batch in an online manner.

# F   Additional results on ImageNet and WMT'17 En-De

In this section we provide additional results for ImageNet and WMT'17 En-De. We also conduct OOD detection using EPKL and RMI as the measures of uncertainty. The results generally confirm the trends observed in the main body of the paper and show that all measures of uncertainty generally behave similarly.

**Image Classification**

As an additional baseline, we have implemented a version of MIMO [25] on ImageNet. To the best of our ability, we recreated the exact setup used in the original work with the differences that we ran it in PyTorch and on GPUs, rather than TPUs. Furthermore, even in the repository, the authors state that their code is TPU-optimized and offer no guarantees for GPUs. Our ImageNet MIMO model has 2 outputs, as was the case in the MIMO paper [25].

The table 8 shows an expanded set of results which includes the MIMO baseline. Note that relative to the MIMO paper ECE is reported as %, not as a fraction. The results demonstrate that on in-domain data (ImageNet-val) MIMO's predictive performance and calibrations are worse than the ensemble and distilled models. However, on shifted data (ImageNet-A/C/R) it has consistently better predictive performance than the distilled models, but overall worse in calibration performance relative to the ensemble and distilled models. This suggests that the MIMO training procedure makes models more robust to distributional shift, but doesn't significantly improve calibration. However, the differences between MIMO, the distilled models (EnD / PD-EnD$^2$) and the ensemble are smaller than the difference between them and a single model.

In addition to assessing predictive quality and calibration of MIMO, we also assess how well it performs on the task of OOD detection. Here we provide results on discriminating between ImageNet Val and ImageNet A/C/R using measures of Total Uncertainty (Entropy of predictive posterior) and Knowledge Uncertainty (Mutual Information). The results show that MIMO performs worse than the ensemble and EnD in all cases except for ImageNet-R using knowledge uncertainty, where it gives the best overall performance. These results show that MIMO is certainly comparable to distribution

distillation, but PD-EnD tends to yield better performance. Note, to the best of our knowledge, these are the first OOD detection results using MIMO — the original paper did not examine this aspect.

Table 9: Out-of-distribution detection results for image classification in terms of % ROC-AUC.

| | ImageNet-O | | | | ImageNet-A | | | | ImageNet-C | | | | ImageNet-R | | | |
|---|---|---|---|---|---|---|---|---|---|---|---|---|---|---|---|---|
| | Total | Knowledge | | | Total | Knowledge | | | Total | Knowledge | | | Total | Knowledge | | |
| | $\mathcal{H}$ | $\mathcal{K}$ | $\mathcal{I}$ | $\mathcal{M}$ | $\mathcal{H}$ | $\mathcal{K}$ | $\mathcal{I}$ | $\mathcal{M}$ | $\mathcal{H}$ | $\mathcal{K}$ | $\mathcal{I}$ | $\mathcal{M}$ | $\mathcal{H}$ | $\mathcal{K}$ | $\mathcal{I}$ | $\mathcal{M}$ |
| Single | 50.7±0.3 | - | - | - | 85.8±0.1 | - | - | - | 79.9±0.4 | - | - | - | 83.0±0.2 | - | - | - |
| MIMO | 49.8 | 58.3 | 57.92 | 58.3 | 86.4 | 85.3 | 85.13 | 85.3 | 79.1 | 81.3 | 81.26 | 81.3 | 83.4 | 87.0 | 86.99 | 87.0 |
| EnD | 48.8±0.6 | - | - | - | 87.2±0.1 | - | - | - | 80.8±0.3 | - | - | - | 83.9±0.2 | - | - | - |
| PD-EnD$^2$ | 52.1±0.2 | 53.2±0.3 | 53.2±0.3 | 53.1±0.3 | 86.8±0.0 | 84.6±0.1 | 84.7±0.1 | 84.4±0.1 | 80.2±0.3 | 77.1±0.4 | 77.4±0.4 | 76.9±0.4 | 83.7±0.1 | 81.7±0.1 | 81.9±0.1 | 81.5±0.1 |
| Ensemble | 54.6 | 62.7 | 61.2 | 62.7 | 88.8 | 86.7 | 87.2 | 86.7 | 82.0 | 77.5 | 78.1 | 77.5 | 86.1 | 84.1 | 84.5 | 84.1 |
| Proxy | 54.6 | 62.7 | 56.2 | 62.9 | 88.8 | 86.7 | 89.0 | 86.5 | 82.0 | 77.5 | 82.1 | 77.3 | 86.1 | 84.1 | 86.3 | 84.0 |

These results show that MIMO is a comparable model in terms of quality, though computational cost of inference is increased somewhat. An advantage of MIMO is that you do not need to train an ensemble first and the do distillation — the cost of training is therefore lower. The primary limitation of MIMO lies in how to generalize it to structured prediction tasks.

**Machine Translation**

Finaly, we also provide a full set of OOD detection results for NMT using additional measures of uncertainty, which generally confirm the trends from section 5.

Table 10: Out-of-distribution detection results for machine translation in terms of % ROC-AUC.

| | Permuted | | | | Speech | | | | German | | | | French | | | |
|---|---|---|---|---|---|---|---|---|---|---|---|---|---|---|---|---|
| | Total | Knowledge | | | Total | Knowledge | | | Total | Knowledge | | | Total | Knowledge | | |
| | $\mathcal{H}$ | $\mathcal{I}$ | $\mathcal{K}$ | $\mathcal{M}$ | $\mathcal{H}$ | $\mathcal{I}$ | $\mathcal{K}$ | $\mathcal{M}$ | $\mathcal{H}$ | $\mathcal{I}$ | $\mathcal{K}$ | $\mathcal{M}$ | $\mathcal{H}$ | $\mathcal{I}$ | $\mathcal{K}$ | $\mathcal{M}$ |
| Single | 80.7±1.5 | - | - | - | 73.7±1.2 | - | - | - | 32.8±2.8 | - | - | - | 27.1±6.3 | - | - | - |
| EnD | 79.5±1.1 | - | - | - | 75.9±0.6 | - | - | - | 35.4±1.6 | - | - | - | 15.6±3.2 | - | - | - |
| EnD$^2$ | 78.3±1.6 | 96.9±0.2 | 96.9±0.3 | 97.1±0.3 | 77.0±0.3 | 76.3±0.2 | 80.4±0.1 | 78.5±0.2 | 38.3±1.6 | 71.4±0.6 | 67.6±1.1 | 70.9±0.7 | 15.9±3.0 | 63.8±3.5 | 51.9±4.1 | 60.1±3.6 |
| Ensemble | 83.7 | 97.2 | 97.4 | 97.4 | 67.8 | 74.4 | 74.0 | 73.7 | 39.5 | 76.3 | 80.2 | 82.4 | 25.0 | 63.6 | 69.9 | 73.6 |