# OpenReview forum: "Scaling Ensemble Distribution Distillation to Many Classes with Proxy Targets"
_NeurIPS.cc/2021/Conference — NeurIPS 2021 Poster_

### Official Review · Reviewer_vef7 · 2021-07-15

**Rating:** 7
**Confidence:** 4

**Summary:**

The authors investigate ensemble distribution distillation for classification models with a high number of classes. This work is based on the work of Malinin et al.[1] which proposed EnD2 to use a Dirichlet prior over categorical predictive distributions to distill ensembles for classification tasks. The authors show through a synthetic example that the gradient norm ratio is very small for increasing number of data classes for the categorical cross-entropy (used in knowledge distillation) and Dirichlet KL (used in EnD2) loss. In contrast, the gradient norm ratio for the reverse KL for Dirichlet distributions maintains similarly for the increasing number of classes. Based on these findings, the author proposes to use a Proxy-Dirichlet target (by calculating the mean and approximating the precision of the ensemble) and use that for the reverse KL objective for ensemble distillation. They evaluate the approach on image classification (ImageNet) and sequence-to-sequence modeling (WMT'17 En-DE). In both settings, the model achieves competitive or better results for performance (Acc, Bleu) and calibration.

**Limitations And Societal Impact:**

This is one of the things that the authors can improve on because they did not discuss limitations and societal impact at all.

**Main Review:**

_Originality & Significance_

This work scales ensemble distribution distillation to a large number of classes which is highly relevant for large-scale modeling. This is very impactful as there are not many works that successfully distilled ensembles of large models and preserves uncertainty. Especially in safety-critical applications, uncertainty estimates are necessary to have some measures of reliability. The large-scale evaluation on ImageNet and WMT show that this approach scales to these datasets/models. This advancement is great for the (research) community.  From the details provided and looking at the supplementary materials, the ensemble distillation for image classification can be reproduced. However, I have doubts about the reproducibility of the auto-regressive model as code has not been provided. There is no discussion about limitations and potential negative impacts which I believe the author should add to make the work complete.

_Quality & Clarity_

Overall, this work is well executed and well-written. The authors give justification why existing knowledge distillation objectives do not scale to data with a large number of classes and propose the reverse KL, which works for these problems. The evaluation is described in detail and results are weighted and discussed. This feels like work very close to completion, and there is not much that I can complain about.

_Comments_

* Can you discuss whether there are alternatives to Sterling's approximation for the Dirichlet precision, how close the approximation comes to the true precision, and whether the approximation might hurt overall distillation?
* Section 3: This is more of an editorial opinion. I found the title "Theoretical analysis" a bit of a stretch. You derive the gradients and the objectives and show through a synthetic example how the gradient norm ratio behaves. I am not sure if this qualifies as theoretical analysis.
* Concurrently, Beyer et al.[2,3] also investigates ensemble knowledge distillation from different points of view. As these two works are relevant and related, it might be worth briefly discussing their works in the part of the related work.
* Can you discuss potential limitations and future directions?

Some typos:
* l.35 "this reduces to KL-divergence" -> this reduces to minimizing
* l.113 "it necessary" -> it is necessary
* l.462 missing reference

[1] Malinin, A., Mlodozeniec, B. and Gales, M., 2019. Ensemble distribution distillation. arXiv preprint arXiv:1905.00076.
[2] Beyer, L., Zhai, X., Royer, A., Markeeva, L., Anil, R. and Kolesnikov, A., 2021. Knowledge distillation: A good teacher is patient and consistent. arXiv preprint arXiv:2106.05237.
[3] Stanton, S., Izmailov, P., Kirichenko, P., Alemi, A.A. and Wilson, A.G., 2021. Does Knowledge Distillation Really Work?. arXiv preprint arXiv:2106.05945.

**Needs Ethics Review:**

Yes

**Time Spent Reviewing:**

4

---

> ### Author Response · Authors · 2021-08-10
> **Reply to Reviewer vef7**
>
> Thank you for your review! Please allow us to answer your questions and address your concerns.
>
> 1. Regarding Stirling’s Approximation. An alternative to Stirling’s approximation would be to match the ensemble’s expected pairwise KL divergence (EPKL), which is a measure of ensemble diversity. For the Dirichlet distribution, EPKL = (K-1)/alpha_0, where K is the number of classes. This approach yields somewhat larger precisions than Stirling’s approximations. However, the resulting distribution-distilled models behaved almost identically. As a result, we decided to go with the more conservative estimates provided by Stirling’s approximation.
>
> * It is in challenging to directly evaluate the quality of the approximation because the ensemble is not Dirichlet distributed and we do not have access to the true precision. However, when comparing the Proxy Dirichlet target to the original ensemble in terms of OOD detection on ImageNet (table 2 and Figures 3c-d), we see that the Proxy target performs comparably to the ensemble - sometimes a little better, sometimes a little worse, but without any obvious degradation.
>
> 2. Regarding Editorial comments. You’re completely right. That section should be called “Analysis of first-order gradients”, rather than “theoretical analysis”, and we will rename it in the next revision.
>
> 3. Regarding concurrent work on analysing distillation. Thanks for pointing these out, they’re very nice papers - we’ll add them as references. We actually noticed something similar, in that the models keep improving as we train for longer (which is why the ASR results were late). In fact, we feel that it takes longer and requires less regularization to do distribution distillation vs. standard distillation. Intuitively this makes sense, as the model is learning more information. The fact that distillation has a far more complex loss surface also explains these observations - presumably, distribution distillation has an even more complex loss surface.
>
> 4. Regarding limitations, negative impacts and future work.
>
> * The main limitation, as we see it, is the long expensive training of distribution distillation. Even with the proposed approach (PD-EnDD), which enables distribution distillation to be applied to large-scale tasks, it still requires constructing an ensemble and then distilling it.
>
> * Future work should therefore investigate how to reduce this cost. Also, I think it would be interesting to do some analysis of the loss surfaces induced by distribution distillation and how it can be done faster and more efficiently in terms of compute.
>
> * With regards to negative impacts - there really are none that we can think of specific to this work, except that this all entails a large carbon footprint, but the same can be said about all of deep learning.
>
> 5. Regarding reproducibility - we will release all the code for this work shortly. Currently, we are cleaning it up and making sure that all the repo is in a presentable state.

---

> > ### Comment · Reviewer_vef7 · 2021-08-23
> > **Reply to the rebuttal**
> >
> > Hi,
> > thanks for the replies to my remaining questions. After reading all reviews and the replies to it, I am keeping my initial score.
> >
> > > 3. Regarding concurrent work on analysing distillation. Thanks for pointing these out, they’re very nice papers - we’ll add them as references. We actually noticed something similar, in that the models keep improving as we train for longer (which is why the ASR results were late). In fact, we feel that it takes longer and requires less regularization to do distribution distillation vs. standard distillation. Intuitively this makes sense, as the model is learning more information. The fact that distillation has a far more complex loss surface also explains these observations - presumably, distribution distillation has an even more complex loss surface.
> >
> > This is an interesting observation! If you can quantify "_less regularization to do distribution distillation vs. standard distillation_", then this may be good to add to the supplementary materials.

---

### Official Review · Reviewer_io27 · 2021-07-15

**Rating:** 6
**Confidence:** 4

**Summary:**

This paper proposes an algorithm for distilling deep ensembles into a single neural network for large-scale classification tasks. The most common baseline in this context would be simply distilling the ensemble models' mean to a single student, but this would throw away some important aspects of ensemble teachers, especially the diversities in predictions. To resolve this, the ensemble distribution distillation (EnD^2) was proposed, where the ensemble predictions are treated as samples from an underlying distribution of distribution (chosen to be Dirichlet), and the student model tries to learn this distribution via maximum likelihood. This paper remedies an important drawback of EnD^2; EnD^2 does not train well for classification tasks with a large number of classes. The authors empirically identify the source of this problem to be the imbalanced gradient norms between the low-probability classes (long-tail classes) and high-probability classes and propose to instead use reverse-KL minimization as objective. To define reverse-KL divergence, the proposed approach constructs a proxy Dirichlet distribution whose parameter is estimated from the ensemble predictions. The proposed EnD^2 variant with proxy Dirichlet target distribution is empirically validated on large-scale image classification and machine translation tasks.

**Limitations And Societal Impact:**

The checklist says that the limitations are described but I find is less apparent. I agree that the content discussed in the paper is unlikely to raise any negative societal impact.

**Main Review:**

Strengths
1. Distilling ensemble models is an important task. As authors pointed out, many baselines either try to improve student networks by increasing the number of parameters (e.g., introducing multi-headed branches). I personally think that the EnD^2 framework is conceptually and theoretically attractive, but empirically found it to be less scalable. The solution provided in this paper can make the EnD^2 framework scale.

2. The experimental results are quite extensive. The paper provides classification results for Imagenet and large-scale machine translation tasks.

3. The paper is well written and easy to follow.

Weaknesses
1. The novelty is somewhat limited. I agree with the author's statement that the proposed reverse KL objective is derived for different purposes (dealing with long-tail classes) compared to the previous work (improving prior networks with reverse-KL objective). Still, the solution provided in this paper is an incremental application of the reverse KL objective (+ proxy Dirichlet ) to the EnD^2 framework.

2. Although the title of section 3 reads as "Theoretical analysis", the actual argument is mainly empirical. The authors analyze the gradient norm ratios between a high-probability class and the others by empirically computing them for three different situations (at initialization, near convergence, and far from convergence) with different losses. Although this analysis has provided enough empirical evidence, it does not provide a deeper understanding of what is going under the hood; for instance, as Figure 1 shows, Dir-RKL with alpha +1 and beta+1 behaves better than the others. Why is this the case? Also, if the gradient ratio rho is affected by the value of the Dirichlet hyperparameters alpha and beta, can Dir-KL or Dir-NLL also be improved by different alpha or beta settings? What about the value of epsilon? In current form, the paper suggests using the reverse KL because empirically it behaved the best under the specific setting considered in section 3. It would be great to have a minimal theoretical analysis (at least with a simplified model) to justify the bad gradient ratio behavior of the Dir-NLL loss and provide a rationale to choose reverse KL instead.

3. The experimental results are extensive but can be strengthened.  Personally, I find evaluating models solely based on ECE to be unreliable., so it would be good to compare the models using various uncertainty assessment metrics such as negative log-likelihoods, MCE, or Brier scores.  Also, for the baselines (especially ensembles or EnD), the calibration errors can further be improved via temperature scaling. I'm not sure whether the temperature scaling is applicable to the proposed method, but if possible it would be best to compare the calibration errors of the baselines and the proposed method under optimally scaled temperatures.

Questions
1. Have you considered using different prior distributions for the classification distribution? For instance, the stick-breaking approximation of the Pitman-Yor process (which is known to better model the long-tail classes than Dirichlet processes), logistic normal distribution, or logistic transform of heavy-tailed distributions. I'm quite skeptical though because judging from the paper EnD^2 suffers from the instability issue because it exaggerates gradients for long-tail classes while the alternative I suggested generally put more mass on long-tail parts of the distributions.

2. In experiments, ensembles of 10 ResNet-50 models are considered for the ImageNet task. How does the proposed method behave with different numbers of ensemble models?  As far as I know, people usually consider five or fewer models for ensembles for ImageNet tasks, because the number of ensembles beyond it is prohibitively expensive. I wonder if the proxy Dirichlet distribution is well constructed with fewer ensemble members.

3. Comparing forward and reverse KL objectives, it is generally known that the forward KL ($\mathrm{KL}[p_\text{target}\Vert p_\text{model}]$) tends to capture the overall mass of the target distribution while the reverse KL ($\mathrm{KL}[p_\text{model}\Vert p_\text{target}]$) tends to capture specific modes. In terms of ensemble distillation, isn't this mode-capturing property of reverse KL bad for capturing diversity?

4. Line 150 states that "if a significant error is made, $\rho$ becomes very large for cross-entropy", but in Figure 1 (c) the $\rho$ values for Cat-KL is similar ($10^0$) with the values for other cases. Can you elaborate on this?

5. Line 195, maybe "do not even begin" rather than "do ven begin"?






**Time Spent Reviewing:**

6 hours

---

> ### Author Response · Authors · 2021-08-10
> **Response to reviewer io27**
>
> Thank you for your review! Please allow us to address the concerns you have raised and the questions you have posed one-by-one.
>
> Regarding Weaknesses:
> 1. Regarding the novelty of the loss. We view the main novelty and contribution of our paper in *enabling* ensemble distribution distillation to be applied to the large-scale tasks of computer vision, machine translation and speech recognition, which could not be done before. In this light, a solution grounded in well-established mathematics whose properties are understood is an advantage, rather than a detriment, in our opinion.
>
> 2. You’re completely right. That section should be called “Analysis of first-order gradients”, rather than “theoretical analysis”.
>
> * There are two reasons for adding +1 to the concentration parameters of both the Dirichlet target and the Dirichlet model. Firstly (high-level explanation), this is done in order to make sure that it does not “invert” (concentration parameter less than 1) along any dimension and has a single, well-defined mode. Inverted Dirichlet’s represent highly undesirable local minima, which are difficult to escape in practice and yield poor gradients. Secondly (low-level explanation),  adding +1 to the digamma functions in the loss prevents them from entering a highly non-linear region where minor perturbations to the concentration parameters yield large changes in the gradient. This improves the stability of optimization.
>
> * Adding +1 to Dirichlet NLL or the Dirichlet Forward KL, while slightly alleviating the gradient problem and preventing inversion, does not resolve the optimization problems entirely.
>
> * As long as epsilon is small, it has very little effect on top of ensuring numerical stability.
>
> * The poor gradient behaviour of the Dir-NLL loss (and the Dirichlet Forward KL)  is simply a property of the loss when optimizing the likelihood of sharp, long-tailed categorical distributions which lie near the edge of the simplex under a Dirichlet distribution, which happens to be the case on real tasks.
>
> * One additional insight we can provide is that when optimizing likelihood or forward KL between categorical distributions, the sum-to-one constraint in the softmax ensures that pulling up the target class probability also pushes down the tail classes, and pushing down the tail classes pushes up the target class. The Dirichlet distribution does not have sum-to-one constraints on the concentration parameters, as they are independent, so the optimization procedure needs to ensure that all classes are correctly modelled. In this situation losses which are highly sensitive to tail classes will have poor convergence properties.
>
>
> 3. Additional calibration metrics. Thank you for the suggestion - we’ve added NLL, ECE, MCE and Brier Score for the ImageNet experiments. Results are here (val is ImageNet validation set, A is ImagnetA, C is ImageNet C and R is ImageNet R):
>
> |          |   val NLL |   val ECE |   val MCE |   val Brier |   A NLL |   A ECE |   A MCE |   A Brier |   C NLL |   C ECE |   C MCE |   C Brier |   R NLL |   R ECE |   R MCE |   R Brier |
> |----------|-----------|-----------|-----------|-------------|---------|---------|---------|-----------|---------|---------|---------|-----------|---------|---------|---------|-----------|
> | Single   |      0.96 |       4.7 |       9.1 |        0.34 |    6.49 |    51.2 |    90.9 |      1.34 |    3.43 |    12   |    24.8 |      0.76 |    4.02 |    20.9 |    34.4 |      0.84 |
> | Ensemble |      **0.81** |       *2.4* |       6.7 |        **0.3**  |    *5.59* |    **42**   |    **88.9** |      **1.23** |    **2.9**  |     **4.6** |     **9.7** |      **0.69** |    *3.48* |     **9.7** |    **15**   |      **0.75** |
> | EnD      |      *0.89* |       **1.6** |       **3.8** |        *0.32* |    5.9  |    46.6 |    90.9 |      1.29 |    *3.09* |     *5.8* |    14.3 |      *0.72* |    3.66 |    16.1 |    26.2 |      0.79 |
> | EnD$^2$    |      *0.89* |       **1.6** |       *4.5* |        *0.32* |    **5.5**  |    *42.8* |    *90.2* |      *1.25* |    3.11 |     **4.6** |    *12.3* |      *0.72* |    **3.39** |    *11.7* |    *20.8* |      *0.77* |
>
> * We can see that the distilled models are always better calibrated than single models. Distribution distilled models often are a little better than standard knowledge-distilled models. We are currently computing the same for translation and speech recognition, but this takes a little longer, as we have to make some modifications to the code and then do inference all over again. Results should be available within the week - we'll post them (and add to the paper) as soon as we have them.
>
> * With regards to temperature calibration - unfortunately, this technique can only be used for unstructured classification, where prediction and calibration do not affect each other. For structured autoregressive models, temperature will directly affect the predictions. Usually higher temperatures result in higher diversity of predictions, but worse performance.
>
> Regarding Questions:
> 1. This is a great question :) . It turns out there are several difficulties with using alternative distributions.
>
> * First, only for the Dirichlet distribution can we easily obtain closed-form expressions for all the different uncertainty measures, as well as the mean and mode of the distribution - other distributions (and indeed processes) would require sampling, which can be very difficult and expensive in certain cases.
>
> * Secondly, the Dirichlet is in some sense the simplest distribution on the simplex, and using a more complex and flexible distribution (such as the logit-normal) would not only require predicting more parameters but, as you point out, would likely make the optimization task even more challenging.
>
> * Finally, it is unclear how non-parametric Processes (Dirichlet, Pitman-Yor) can be learned via gradient-based optimization.
>
> 2. In general, bigger and better ensembles will yield better proxy targets, both in terms of robustness of estimation and in terms of the performance upper bound they specify. Furthermore, the quality of the proxy doesn’t impact our ability to optimize the loss, assuming it is specified as described in the paper.
>
> * The mean of the proxy Dirichlet is always well estimated and equal to the ensemble mean (by construction). The estimation of the precision, obtained via Stirling’s approximation, may have high variance if estimated using very few points. However, deep ensembles typically begin to show diminishing returns at the 4-6 model point [1], with 10 already saturating, so we suspect that a fairly representative estimate of the precision can be obtained from 4-6 models.
>
> [1] Uncertainty Estimation in Autoregressive Structured Prediction, Malinin and Gales, 2021
>
> 3. Forward and Reverse KL will drive the model to different solutions if the target and approximating distributions are of a different type. For example, if the target distribution (P) is a Bimodal mixture of Normal distributions and the approximating distribution (Q) is a unimodal normal distribution, then Forward KL will yield *mean-matching* behaviour, where the unimodal normal Q will spread itself between the two modes of the target P. Reverse KL, will yield mode-seeking behaviour where the unimodal model Q will pick one mode of P and sit on it. Additionally, in this case, neither FKL nor RKL can ever reach 0.
>
> * In our case, this isn’t a problem, as both the model and the target (Proxy Dirichlet) are of the same type (Dirichlet distribution).  This means that Forward KL and Reverse KL have the same global solution - loss is 0 when the model == Proxy Dirichlet target. However, it is necessary to point out that they yield different behaviour with respect to gradient-based optimization, which is the problem at the heart of this paper.
>
> 4. Thanks for finding this! The sentence contains an error, what we meant to say was:  “...if a significant error is made, $\rho$ remains large for cross-entropy and Dirichlet RKL, but becomes small for Dirichlet NLL Forward KL..."  We meant to highlight that when an error occurs, the desired property to maintain is a high $\rho$, but $\rho$ for Dir-NLL/FKL drops down and the optimization begins focus on the tail again.

---

> > ### Comment · Reviewer_io27 · 2021-08-11
> > **Thanks for the rebuttal!**
> >
> > Dear authors, thanks for the detailed response. Your response resolved most of my concerns, especially the discussion of the rationale for using reverse KL divergence, and additional results with uncertainty evaluating metrics. Thus I raise my score to 6.

---

### Official Review · Reviewer_eKFR · 2021-07-16

**Rating:** 8
**Confidence:** 5

**Summary:**

This paper considers uncertainty estimation for neural networks, specifically predictive uncertainty and knowledge uncertainty. For these problems, deep ensembles are considered state-of-the-art, but are expensive at training time and, more importantly, inference time. To address the latter problem, this paper explores Ensemble Distribution Distillation (EnD^2), an approach previously proposed by Malinin et al (2020) which consists of distilling the ensemble into a single model that captures both the mean and diversity of the original ensemble. However, this paper notes that EnD^2 fails to converge for problems where there is a large number of classes, which is attributed to the use of the forward KL divergence as the distillation criterion. Instead, the reverse-KL divergence is proposed using a proxy Dirichlet distribution to represent the ensemble. The precision of the proxy distribution is estimated using Stirling's approximation, and some constraints are imposed on the model parameters to ensure stability. Experiments on ImageNet (several variants) and NMT show that the proposed approach successfully retrains some of the accuracy and calibration benefits of the corresponding deep ensemble.





**Limitations And Societal Impact:**

Yes.

**Main Review:**

Overall, I found the paper to be well-written, with a reasonable motivation for the main contribution, PD-EnD^2, and a convincing analysis and explanation of the proposed approach. To the extent that vanilla EnD^2 *does not work* on the class of problems considered in this work, and with distillation a requirement for deep ensembles to be practical in many settings, this paper fills an important gap in the literature.

That being said, the experimental comparisons were somewhat lacking, as discussed further below. Notably, MIMO has all of the benefits of the proposed approach (albeit with several disadvantages), which make it an interesting baseline. In addition, there is related work using (vanilla) ensemble distillation for NMT that may warrant experimental comparison, given that the sequence-level extension of PD-EnD2 is a non-trivial contribution of this work.

It is stated that "we do not provide results for EnD2" because "these approaches do even begin to converge on the tasks considered." I think it would have improved the submission to characterize the point at which vanilla EnD2 starts to fail, for example by varying the number of classes, or even using a simple synthetic example. It would be interesting to know, for problems with fewer classes, whether PD-EnD^2 always outperforms vanilla EnD^2.

There are a couple notable missing references:

* The paper mentions the alternate approach of capturing ensemble diversity using separate output heads for each ensemble member, but doesn't cite MIMO (https://arxiv.org/pdf/2010.06610.pdf). Notably, MIMO has similar computational cost for small ensembles as a single network, and each network-specific head only introduces a small amount of parameters. Therefore, I don't think the objections raised apply to MIMO, and it would have been nice to see experimental comparisons against MIMO on ImageNet (https://github.com/google/uncertainty-baselines).

* Ensemble distillation (vanilla, not EnD2) has been applied to NMT (https://arxiv.org/abs/2010.06721), where a truncation is proposed to avoid modeling the long-tail. This has the practical implication of allowing for caching of the predictive distribution of the ensemble, obviating the need to hold all ensemble members in memory as is done in this work (which compensates for it with 32 steps of gradient accumulation). In the context of the EnD2 approach used here, the truncation could have another benefit by helping the Dirichlet NLL objective avoid trying to perfectly model the long-tail, and therefore may merit some consideration / discussion. In addition, the reported results could be compared to those in that work.

** Update after author response ** The authors have addressed several of these concerns with additional experiments, so I am revising my recommendation upwards.

Minor issues

* L185: missing "for"

**Time Spent Reviewing:**

3

---

> ### Author Response · Authors · 2021-08-10
> **Response to Reviewer eKFR**
>
> Dear Reviewer!
>
> Thank you for your review! Please let us address your concerns regarding the thoroughness of our experimental evaluation. Before we address your points one-by-one, please allow us to highlight that in addition to the image-classification and machine translation results in the main paper, we also provide results for autoregressive speech recognition on LibriSpeech, where PD-SEnD$^2$ also works. These results are presented in the appendices only because the full set of experiments converged a day after the main paper deadline, and will be moved to the main body in future revisions of the paper.
>
> 1. Thank you for suggesting a synthetic experiment. In the original paper on Ensemble Distribution Distillation (Malinin et al) a synthetic experiment with 3 classes (spiral dataset) was run which demonstrated EnD$^2$. However, on the image classification experiments a number of temperature-based heuristics were used to improve convergence, which suggests that the authors had already encountered optimization difficulties at the level of 10 classes. Interestingly, this generally agrees with our analysis in figure 1, where we show that rho is larger for Dirichlet RKL than for Dirichlet NLL for any number of classes.
>
> * We’ve decided to run an extension of the synthetic spiral experiment from the original Ensemble Distribution Distillation paper. We will train ensembles of 50 feed-forward classification DNNs and vary the number of spirals (classes) from 3 to 100 (3, 10, 50, 100), and compare how vanilla EnD$^2$ fares against PD-EnDD. Note, that we will not use the additional temperature tricks for ‘vanilla EnDD’ in order to have a clean comparison. The results will be added to the appendix. We will post the results here as soon as they are ready.
>
> 2. Regarding MIMO. Thank you for pointing out this work. We are currently spinning up a MIMO baseline for ImageNet which we will include in the paper. We expect results within this week.
>
> * Note, however, that we are unable to replicate MIMO for Machine Translation and Speech recognition because the extension is highly non-trivial. For truly independent prediction of several variable-length sequences given several variable-length inputs, it would require a concatenation of both the encoder and the decoder inputs in some fashion. If we concatenate in time, this would lead to a significant (quadratic) growth in computation and memory costs for Transformers than for image classification models (where inputs can be easily concatenated). Concatenating along the embedding dimension would require figuring out how to concatenate variable-length inputs and decode multiple variable-length outputs.
>
> * An advantage of EnD$^2$ over MIMO is that it can be implemented both for feed-forward image classification models and for autoregressive translation / speech recognition models almost without modification and with truly no growth in the amount of computation relative to a single model.
>
> 3. Regarding truncation of the softmax. Thanks for the paper, we’ll cite it!
>
> * We actually tried something like this at the very beginning of our work for machine translation. We tried to "virtually" truncate the distribution for each input to the top-10 most likely classes (computed dynamically for each softmax), and contain the probability mass of the long tail in a virtual 11-th “the rest” class. We then used the aggregation property of the Dirichlet Distribution to reduce the dimensionality of the problem by summing the predicted concentration parameters (alpha_k) for the *ensemble’s* long-tail classes into a single virtual class which models the bulk behaviour of the long tail and from which the gradients can propagate to the alphas of the long-tail classes. When combined with very large batches, this alleviated the gradient issues and allowed us to train a model using the Dirichlet NLL loss function up to 28 BLEU, but still less than a single 'normal' model. Also, this model had very poor measures of uncertainty.
>
> * We also tried aggregation in conjunction with the Proxy Dirichlet and *forward* KL divergence. This also alleviated the gradient issues somewhat, but the models still underperformed a single model and had uninformative uncertainties. Despite months of effort, we were unable to get this approache to yield anything better (we tried various smoothing approaches, different parameterizations, alternative optimization strategies, etc…)  and dropped this approach in favour of the Proxy Dirichlet + RKL approach. We found that aggregation was unnecessary if using Proxy Dirichlet in conjunction with the reverse KL-divergence loss as presented in the paper, which is why we didn’t mention it. We can add a section in the appendix describing some of these negative results.

---

> > ### Comment · Reviewer_eKFR · 2021-08-19
> > **Thanks for the detailed reply!**
> >
> > Thanks for the comprehensive response! I look forward to seeing the synthetic experiments and the comparisons with MIMO (on ImageNet). Regarding truncation, I'm somewhat surprised to hear that it was unsuccessful considering it seems like a simple & efficient way to (approximately) account for the long-tail. Therefore, I think the paper could be improved by including these results as a baseline in the main text rather than in the appendix, as other readers may also wonder about this alternative.

---

> > > ### Author Response · Authors · 2021-08-23
> > > **Response regarding truncation**
> > >
> > > Thank you for your comments. We have provided a comprehensive set of additional results in the general response above.
> > >
> > > Regarding truncation, I believe there are two parallel stories here - one about computational cost, and the other about overcoming the gradient issues of EnD$^2$. Let's separately examine both.
> > >
> > > 1. Truncation for overcoming gradient issues.
> > >
> > > Our initial experiments showed that via Aggregation of tail classes into a single class, which is a form of "virtual truncation" for loss computation, can help alleviate the gradient problem. However, this *does not* resolve the issue. Furthermore, aggregation down to 10+1 classes was still insufficient. As we can see both from the plots of $\rho$ vs number of classes and from the new synthetic experiments - the gradient issues persist even for 10 or fewer classes. Thus, we would need to truncate down to 3 or so classes to get any form of sensible convergence. However, for tasks with many classes, such as translation and speech recognition, 3 classes may be too great of an approximation and lots of valuable information about related words/tokens will be lost. We observed that roughly 90% of the probability mass tends to be contained within the top 10 - 100 classes, depending on the input. Thus, truncating down to 3 +1 or 2+1 classes will lead to loss of information. Furthermore, the top-1 class may cease to be the top 1 class after aggregating so much probability mass to the +1 "tail" class.
> > >
> > > Thus, aggregation, and presumably other forms of truncation, will not be able to resolve the gradient issues of the Dirichlet NLL loss described in the paper - only alleviate them to some degree. On the other hand, the proposed solution does not require any form of truncation to allow models to converge to the desired behaviour for any task (ImageNet / NMT / ASR).
> > >
> > >
> > > 2. Truncation for computational savings
> > >
> > > While the proposed solution (PD-EnD$^2$ ) does not *require* truncation/aggregation to converge to good solutions, we *can* still use aggregation to reduce the computational and memory cost - the aggregation technique we examined is orthogonal to and compatible with PD-EnD$^$. Presumably, cutting down the number of probabilities stored from 40K down to 100+1 would provide good fidelity of modelling (as this would cover around 90% of the probability mass), and also reduce the memory cost by about 2 orders of magnitude.

---

### Review · Ethics_Reviewer_9vEV · 2021-07-23

**Recommendation:**

The authors could further discussed potential impacts. However, the paper does not raise ethical concerns in general.

**Ethics Review:**

The paper could do more to discuss potential impacts, as no discussion is included currently. However, the paper does not raise ethical concerns.

The authors point out that "uncertainty estimation has no concievable negative societal impact." It's plausible miscalibrated uncertanties could lead to overconfidence in a model, leading to potential harms. This could serve as a potential idea to include.

---

> ### Author Response · Authors · 2021-08-23
> **Response to Ethics Reviewer 9vEV**
>
> Thank you for your comments!
>
> What we meant by that line was that work on *improving* uncertainty estimation, calibration and robustness doesn't have any conceivable negative societal consequences unlike, for example, work on DeepFakes or Language Models, which can be abused. We will clarify this point and add a more expanded discussion of the consequences of poor calibration in addition to the modifications which we promised reviewer ver7.

---

### Review · Ethics_Reviewer_N2Ls · 2021-08-12

**Recommendation:** N/A

**Ethics Review:**

The paper addresses the problem of deriving ensembles of machine learning models without prohibatively expensive inference costs. The author improve upon the Ensemble Distribution Distillation (EnD2) by proposing a new training objective to resolve graident issues of EnD2.

Overall, the paper does not appear to have unaddressed potential negative social impacts. In their response to Reviewer vef7, the authors clearly list potential limitations and societal impacts of the model. Given that the contribution is mainly methodological, the limitations, negative impacts, and future work section is sufficient. The analysis appears to follow general ethical conduct.

Therefore I do not raise any ethical concerns that need to be addressed.

---

### Author Response · Authors · 2021-08-11
**Concise summary of changes.**

General Response

Dear Reviewers, thank you for your detailed reviews and constructive feedback.

In addition to answering and addressing your concerns on a point-by-point basis, we would also like to provide a concise overall summary of the changes we will make. Although all reviewers agreed that the problem is well motivated and the paper is well written, reviewers eKFR and io27 raised some concerns regarding the thoroughness of the experimental evaluation, which we are addressing as follows:

1.  Please allow us to highlight that in addition to the image classification and machine translation results, in the appendix we also provide a set of results demonstrating the PD-SEnD$^2$ works for autoregressive speech recognition trained on LibriSpeech as well. These results were presented in the supplementary because the full set of experiments converged a day after the main paper deadline. In future revisions of this paper, we will move these results to the main body of the paper.

2. As we have replied to reviewer eKFR, we are currently spinning up a MIMO baseline for ImageNet. Unfortunately, we can do this only for image classification, as extending MIMO to autoregressive models is non-trivial.

3. We are also spinning up a synthetic experiment that compares vanilla EnD$^2$ vs PD-EnD$^2$ on a synthetic task as the number of classes is increased.

4. We are providing an expanded set of calibration performance metrics for all tasks. Metrics have been obtained for ImageNet (presented in response to io27) and we are currently calculating them for machine translation and speech recognition. So far the results consistently show that distilled models are better calibrated than baseline single models.

We expect that all additional results will be ready this week.

---

> ### Author Response · Authors · 2021-08-23
> **Additional experimental results pt 1**
>
> 1. Synthetic Results
>
> We ran the synthetic experiment on the spiral dataset with 3, 10 and 20 classes (50 was unnecessary). Identical hyperparameters and models were used for both EnD$^2$ and PD-EnD$^2$ . The only difference was the addition of 1 Layernorm layer for PD-EnD$^2$, which helped *numerically* stabilize training.
>
> Regarding stability - the spiral dataset is such that the variation in ensemble precision across different input locations can be absolutely enormous - rare points right on the arms of the spiral can be several orders of magnitude larger precisions than in other locations. This leads to numerical instabilities and NAN occasionally in PD-EnD$^2$ due to division by precision. We resolved this by using a single LayerNorm layer just before the final output layer. Using LayerNorm (or similar) for standard EnD$^2$ via Dirichlet NLL prevented convergence altogether and was not used. We suspect that a more numerically stable implementation of the loss would not require LayerNorm. Note that adding LayerNorm did not improve the results for PD-EnD$^2$ - it only prevented any training run from aborting due to NANs in the loss.
>
> The results show that on 3 classes we can more or less match the error rate of an ensemble/single model with both EnD$^2$ and PD-EnD$^2$, thought standard EnD$^2$ displays greater variation. As the number of classes is increased, the PD-EnD$^2$ consistently yields low error rates. Standard EnD$^2$ displays growing error rates and failure of convergence for 20 classes.
>
> Note, we explicitly did not use the heuristic temperature tricks from the original paper in order to compare pure forms of the Dirichlet Likelihood of PD-EnD$^2$ reverse KL objectives. The results generally agree with the story presented in the plots of $\rho$ vs the number of classes where it is clear that even for 10 classes $\rho$ for reverse KL loss is larger than for the Dirichlet-NLL loss.
>
> Error Rates on heldout data
>
>
> |               |   3 | 6 |  10   |   20 |
> |--------------|--------|--------|--------|--------|
> | Single       |  1.48$_{\pm 0.36}$  | 1.93$_{\pm 0.39}$ |   0.99$_{\pm 0.49}$    | 0.75$_{\pm 0.24}$  |
> | EnD$^2$         |   1.98$_{\pm 0.49}$ | 2.38$_{\pm 0.49}$   |   4.92$_{\pm 2.12}$     |  -   |
> | PD-EnD$^2$        |  1.22$_{\pm 0.11}$  |  1.57$_{\pm 0.09}$ |  0.61$_{\pm 0.03}$ |   0.52$_{\pm 0.06}$ |
> | Ensemble       | 1.1 | 1.6 |  0.5 |  0.5 |
>
> Additionally, we examined the models' median precisions and compared to the ensemble's (obtained via Stirling). We used median over the heldout dataset as the presence of rare inputs points right at the center of the spiral arms, whose precisions were orders of magnitude larger greatly affected the mean. The results show that standard EnD$^2$ yields low precisions which decrease as the number of classes grows, while PD-EnD$^2$ consistently yields precisions comparable to the ensemble.
>
> Model Precision
>
> |               |   3 | 6 |  10   |   20 |
> |--------------|--------|--------|--------|--------|
> | EnD          |  278.7$_{\pm 69.1}$  |  13.6$_{\pm 4.8}$ | 11.75$_{\pm 0.61}$    |  -   |
> | EnD$^2$        |  4413.5$_{\pm 489.3}$   |  2791.2$_{\pm 148.0}$  | 2691.5$_{\pm 106.4}$ | 3985.2$_{\pm 187.4}$ |
> | Ensemble       |   14350.7 |  10275.9 | 8859.8 | 6049.4  |
>
> Finally, we examined heat-maps of uncertainty values across a fine-grid of inputs and observed a degradation for K > 3 for standard EnD$^2$, while PD-EnD$^2 produced behaviour similar to that of the ensemble. Unfortunately we cannot attach them in any way to OpenReview.
>
> These results generally agree with the plots of $\rho$ vs number of classes, which show that for standard EnD$^2$ (Dirichet NLL) there is an immediate drop in $\rho$, while for the proposed approach $\rho$ remains high.
>
>
> 2. MIMO Results
>
> We implemented a version of MIMO on ImageNet. To the best of our ability, we recreated the exact setup used in the original work with the differences that we ran it in PyTorch and on GPUs, rather than TPUs. Furthermore, even in the repo, the authors state that their code is TPU-optimized and offer no guarantees for GPUs.
>
> Our ImageNet MIMO model has 2 outputs, as was the case in the MIMO paper.
>
> The table below shows an expanded set of results which includes the MIMO baseline. Note that relative to the MIMO paper ECE is reported as %, not as a fraction. The results demonstrate that on in-domain data (ImageNet val) MIMO's predictive performance and calibrations are worse than the ensemble and distilled models. However, on shifted data (ImageNet A/C/R) it has consistently better predictive performance than the distilled models, but overall worse in calibration performance relative to the ensemble and distilled models. This suggests that the MIMO training procedure makes models more robust to distributional shift, but doesn't significantly improve calibration. However, the differences between MIMO, the distilled models (EnD / PD-End$^2$) and the ensemble are smaller than the difference between them and a single model.
>
>
> |              |   val Acc |   val NLL |   val ECE |   val MCE |   val Brier |   A Acc |   A NLL |   A ECE |   A MCE |   A Brier |   C Acc |   C NLL |   C ECE |   C MCE |   C Brier |   R Acc |   R NLL |   R ECE |   R MCE |   R Brier |
> |--------------|-----------|-----------|-----------|-----------|-------------|---------|---------|---------|---------|-----------|---------|---------|---------|---------|-----------|---------|---------|---------|---------|-----------|
> | Single       |      76   |      0.96 |       4.7 |       9.1 |        0.34 |     4.3 |    6.49 |    51.2 |    90.9 |      1.34 |    38.6 |    3.43 |    12   |    24.8 |      0.76 |    34.7 |    4.02 |    20.9 |    34.4 |      0.84 |
> | Ensemble     |      79   |      0.81 |       2.4 |       6.7 |        0.3  |     3.9 |    5.59 |    42   |    88.9 |      1.23 |    43.5 |    2.9  |     4.6 |     9.7 |      0.69 |    38.8 |    3.48 |     9.7 |    15   |      0.75 |
> | MIMO ($M=2$) |      76.8 |      0.91 |       2.7 |       4.7 |        0.32 |     5.2 |    5.9  |    47.4 |    90   |      1.29 |    41.3 |    3.05 |     8.1 |    18.2 |      0.72 |    37.7 |    3.67 |    17.6 |    27.7 |      0.79 |
> | EnD          |      77   |      0.89 |       1.6 |       3.8 |        0.32 |     3.9 |    5.9  |    46.6 |    90.9 |      1.29 |    40.4 |    3.09 |     5.8 |    14.3 |      0.72 |    36.8 |    3.66 |    16.1 |    26.2 |      0.79 |
> | EnD$^2$       |      77.1 |      0.89 |       1.6 |       4.5 |        0.32 |     4   |    5.5  |    42.8 |    90.2 |      1.25 |    40.2 |    3.11 |     4.6 |    12.3 |      0.72 |    36.9 |    3.39 |    11.7 |    20.8 |      0.77 |
>
> In addition to assessing predictive quality and calibration of MIMO, we also assess how well it performs on the task of OOD detection. Here we provide results on discriminating between ImageNet Val and ImageNet A/C/R using measures of Total Uncertainty (Entropy of predictive posterior) and Knowledge Uncertainty (Mutual Information). The results show that MIMO performs worse than the ensemble and than EnD$^2$ in all cases except for ImageNet-R using knowledge uncertainty, where it gives the best overall performance. These results show that MIMO is certainly comparable to distribution distillation, but PD-EnD$^2$ tends to yield better performance. Note, to the best of our knowledge, these are the first OOD detection results using MIMO - the original paper did not examine this aspect.
>
> |              |   A TU | A KU   |   C TU | C KU   |   R TU | R KU   |
> |--------------|--------|--------|--------|--------|--------|--------|
> | Single       |  85.86 | -      |  79.77 | -      |  83.29 | -      |
> | Ensemble     |  88.8  | 87.21  |  82.19 | 78.36  |  86.12 | 84.54  |
> | MIMO ($M=2$) |  86.37 | 85.13  |  79.13 | 81.26  |  83.44 | 86.99  |
> | EnD          |  87.12 | -      |  80.95 | -      |  83.89 | -      |
> | PD-EnD$^2$        |  87.52 | 86.75  |  81.56 | 81.48  |  84.81 | 85.42  |
>
> These results show that MIMO is a comparable model in terms of quality, though computational cost of inference is increased somewhat. An advantage of MIMO is that you don't need to train an ensemble first and the do distillation - the cost of training is therefore lower. The primary limitation of MIMO lies in how to generalize it to structured prediction tasks.

---

> > ### Author Response · Authors · 2021-08-23
> > **Additional results pt2**
> >
> > 3. Remaining calibration results
> >
> > As we promised reviewer io27, we now provide NLL results for NMT. Note, that NLL results for ASR are provided in the supplementary. Before we begin discussing the results, we would like to highlight that calibration for classification tasks and calibration for structured prediction have different attributes. In the latter case calibration strongly interacts with predictions and the context in the decoder.
> >
> > In the table below we proide BLEU and NLL (average length-normalized sentence NLL) relative to both REFERENCE transcriptions and also to the ENSEMBLE'S transcriptions. We evaluate this on Newstest'17 (Test). The results show that the ensemble has better (lower) NLL than a single model on reference transcriptions, but both distilled models have worse (higher) NLL than both the ensemble and single model on REFERENCE transcriptions. However, these models were never trained on the reference data - they were always trained on transcriptions provided *by the ensemble*. Thus, on ENSEMBLE transcriptions, both distilled models have better NLLs than the single model and higher BLEU scores relative to the ensemble. Also note that the NLLs of ALL models are better (lower) on ensemble predictions than on the references. This is consistent with the observations made in [1] that references transcriptions are more complex and multimodal, more surprising, therefore would have higher NLLs,  while predictions of NMT models are more consistent in style and therefore simpler - they are less 'surprising', and therefore have lower NLL.
> >
> > In short, these results show that while all distilled models now consider the references to be more surprising, as their decoders were never exposed to reference text, the distilled models are more closely matched to the ensemble. Furthermore, despite seemingly poorer calibration on reference text, the distilled models nevertheless have better predictive performance. These results all suggest that the interaction between calibration and predictive quality in structured prediction models is a fruitful area of research.
> >
> > |          |   Test BLEU |   Test NLL |   Test (ens pred) BLEU |   Test (ens pred) NLL
> > |----------|-------------|------------|------------------------|-----------------------|
> > | Single   |       28.8  |       1.46 |                  71.54 |                  0.42 |
> > | Ensemble |       30.13 |       1.33 |                  99.88 |                  0.39 |
> > | SEnD      |       29.45 |       1.52 |                  77.73 |                  0.4  |
> > | PD-EnD$^2$    |       29.47 |       1.52 |                  77.31 |                  0.4  |
> >
> >
> > [1] "Understanding Knowledge Distillation in Non-autoregressive Machine Translation". Chunting Zhou, Jiatao Gu, Graham Neubig

---

> > ### Comment · Reviewer_eKFR · 2021-08-30
> > **Thanks for the additional results!**
> >
> > These results address some concerns raised in my original review so I will revise my recommendation accordingly.

---

### Decision · Program_Chairs · 2021-09-27

**Decision:**

Accept (Poster)

**Comment:**

The reviewers are all for accepting the work. It's easy to follow and has quite extensive experiments, especially with the help of the MIMO baseline in the rebuttal (please cite and add the additional experiments into the paper). The loss function involves several moving parts, as well as pipelines like distilling from a trained model with much longer training than the baselines they compare to. So the idea isn't necessarily easy to implement. With that said, I think the work's ideas and its empirical evaluation will be of interest to the community.